# HiMAP-Travel: Hierarchical Multi-Agent Planning for Long-Horizon Constrained Travel

**The Viet Bui** [* 1]  **Wenjun Li** [* 1]  **Yong Liu** [2]

## Abstract

Sequential LLM agents fail on long-horizon planning with hard constraints like budgets and diversity requirements. As planning progresses and context grows, these agents drift from global constraints. We propose **HiMAP-Travel**, a hierarchical multi-agent framework that splits planning into strategic coordination and parallel day-level execution. A Coordinator allocates resources across days, while Day Executors plan independently in parallel. Three key mechanisms enable this: a transactional monitor enforcing budget and uniqueness constraints across parallel agents, a bargaining protocol allowing agents to reject infeasible sub-goals and trigger re-planning, and a single policy trained with GRPO that powers all agents through role conditioning. On TravelPlanner, HiMAP-Travel with Qwen3-8B achieves 52.78% validation and 52.65% test Final Pass Rate (FPR). In a controlled comparison with identical model, training, and tools, it outperforms the sequential DeepTravel baseline by +8.67 pp. It also surpasses ATLAS by +17.65 pp and MTP by +10.0 pp. On FlexTravelBench multi-turn scenarios, it achieves 44.34% (2-turn) and 37.42% (3-turn) FPR while reducing latency 2.5× through parallelization.

## 1. Introduction

The rapid ascent of Large Language Models (LLMs) has redefined the landscape of autonomous agents, enabling systems that synthesize vast knowledge bases into coherent, actionable plans. However, a critical gap remains between their proficiency in short-horizon tasks and their fragility in

long-horizon, logically constrained planning. While modern LLMs excel at open-ended generation, they degrade sharply on combinatorial optimization problems that require the simultaneous satisfaction of rigid hard constraints–such as strict budgets, temporal feasibility, and route consistency–alongside nuanced user preferences. Autonomous travel planning serves as a rigorous testbed for this challenge, where a single budget violation or logistical error on the first day can invalidate an otherwise optimal multi-day itinerary.

The prevailing paradigm for these tasks relies on monolithic, sequential architectures, typified by frameworks such as ReAct (Yao et al., 2022) or Chain-of-Thought (CoT; (Wei et al., 2022)) prompting, in which a single policy generates the entire trajectory token-by-token. We posit that this approach suffers from a failure mode we term *Constraint Drift under Long Tool Traces*. As the planning horizon extends, intermediate tool outputs, search logs, and reasoning traces accumulate, increasing the effective context length and diluting attention to the initial global constraints (e.g., the total budget specified in $q$). This produces a measurable decay in global feasibility and exposes a structural limitation of sequential planning: strategic resource allocation and tactical execution are entangled in one growing context.

Verify-and-refine methods such as Reflexion (Shinn et al., 2023) and ATLAS (Choi et al., 2025) mitigate such errors through post-hoc checking, but this repair loop becomes increasingly costly as agentic tasks scale. A 7-day itinerary may be generated in full before a Day 1 budget error is detected, wasting computation and increasing latency. A scalable solution requires *proactive* constraint enforcement during generation, not only retroactive correction. By parallelizing execution and enforcing constraints atomically via a shared state, we shift the paradigm from "generate-then-fix" to "correct-by-construction."

To surmount these architectural limitations, we propose **HiMAP-Travel** (**Hi**erarchical **M**ulti-**A**gent **P**lanning for Travel), a framework for long-horizon travel itinerary generation. Our core insight is to structurally decouple the optimization landscape into two distinct functional levels: a *Strategic Level* managed by a high-level *Coordinator*, and a *Tactical Level* managed by decentralized *Executors*. The Coordinator projects the high-dimensional user query into

---

[*]Equal contribution  [1]School of Computing and Information Systems, Singapore Management University, Singapore  [2]Independent Researcher. Correspondence to: Wenjun Li <wenjunli2017@gmail.com>.

*Proceedings of the 43rd International Conference on Machine Learning*, Seoul, South Korea. PMLR 306, 2026. Copyright 2026 by the author(s).

a set of valid local boundary conditions (sub-goals), effectively distributing global constraints–such as total budget and temporal duration–across a team of specialized agents. Crucially, rather than operating as a rigid command-and-control hierarchy, we implement a *Cooperative Bargaining Protocol* where Executors can reject infeasible sub-goals and provide structured feedback, enabling the Coordinator to dynamically re-allocate resources until convergence. In our implementation, the Coordinator provides *soft* per-day guidance (e.g., budget hints and roles), while coupled *hard* constraints (e.g., a shared global budget and global non-duplication) are enforced deterministically by a synchronized monitor during execution. This decomposition allows Executors to operate in parallel within isolated, pristine context windows, ensuring that the planner for Day $N$ is independent of the noise generated by the execution of Day 1. By factoring the state space, we reduce the effective context length per subproblem from $\mathcal{O}(T)$ to $\mathcal{O}(T/D)$, effectively mitigating the effects of Constraint Drift.

**Key System Contribution.** While hierarchical planning is an established paradigm, HiMAP-Travel differentiates itself through a **Synchronized Parallel Architecture** that enforces coupled constraints *by construction* rather than post-hoc verification. Unlike methods that rely on "Constraint Managers" to critique full plans (e.g., ATLAS), our contribution is the synthesis of three integrated mechanisms: (1) a **Transactional Constraint Monitor** that provides deterministic safety guarantees for parallel agents via atomic locks, preventing resource conflicts before they occur; (2) a **Cooperative Bargaining Protocol** that enables dynamic re-planning via structured signals rather than verbose dialogue; and (3) a **Unified Role-Conditioned Policy** trained via GRPO. This integration allows us to outperform sequential baselines using the *same* underlying LLM and environment tools, isolating the architectural benefit.

Our contributions are fourfold. First, we propose **HiMAP-Travel**, a **Hierarchical Multi-Agent** framework that decouples planning into a Coordinator for strategic resource allocation and parallel Executors for tactical resolution, significantly reducing sequential error propagation. Second, we introduce a **Synchronized Global State**, a deterministic transactional monitor that enforces coupled constraints (e.g., budget, diversity) across parallel agents without natural language negotiation overhead. Third, we implement **Cooperative Bargaining**, a structured feedback loop where Executors can reject infeasible sub-goals, enabling recovery from initial allocation errors. Finally, we achieve **State-of-the-Art Performance** on the TravelPlanner benchmark, with empirical results showing 52.78% validation and 52.65% test Final Pass Rate, surpassing sequential baselines while reducing latency by 2.5×.

**Conflict of Interest Disclosure.** The authors declare no financial or other substantive conflicts of interest relevant to this work.

## 2. Related Work

**Tool-Augmented Agentic Planning.** The paradigm of LLM agents has evolved rapidly from simple few-shot prompting to sophisticated closed-loop architectures. Foundational frameworks like ReAct (Yao et al., 2022) and Toolformer (Schick et al., 2023) established the viability of interleaving reasoning traces with external API calls. However, monolithic sequential planners scale poorly to long horizons: as tool outputs and reasoning traces accumulate, attention to initial global constraints degrades–a phenomenon we term *Constraint Drift*. Recent mitigation strategies like ATLAS (Choi et al., 2025) employ a multi-agent *verify-and-refine* loop with a Constraint Manager and interleaved search, reporting 35% test Final Pass Rate (FPR) on TravelPlanner with Gemini-2.5-Pro. Meta-Task Planning (MTP; (Zhang et al., 2024)) introduces a structured hierarchy of agents (Manager/Executor/Supervisor) and achieves 42.68% Test FPR with GPT-4 when given unconventional hints. However, both methods suffer from super-linear latency scaling due to post-hoc correction mechanisms. In contrast, HiMAP-Travel enforces constraints proactively during generation and parallelizes day-level planning to reduce long-horizon brittleness and latency.

**Autonomous Travel Planning and Benchmarking.** Autonomous travel planning is a rigorous testbed for evaluating agents under coupled logical constraints (Gavalas et al., 2014). Benchmarks such as TravelPlanner (Xie et al., 2024) evaluate itinerary generation under shared invariants (e.g., budget, diversity, non-duplication), while FlexTravelBench (Oh et al., 2025) introduces progressive constraint revelation to test multi-turn adaptation. Prior methods such as DeepTravel (Ning et al., 2025) apply end-to-end reinforcement learning (RL) to optimize tool usage, but generates itineraries sequentially, so early sub-optimal decisions (e.g., overspending on Day 1) cascade downstream and limit performance on high-complexity queries. HiMAP-Travel mitigates this sequential dependency via day-level decomposition with context isolation, and uses Cooperative Bargaining to re-allocate resources when a day plan is infeasible or constraints evolve.

**Hierarchical and Multi-Agent Reasoning.** Hierarchical Reinforcement Learning (HRL) addresses the curse of dimensionality by separating goal setting from control (Sutton et al., 1999), and recent systems adapt hierarchical decomposition for planning (e.g., retrieval-based approaches such as HiPlan (Li et al., 2025)). Multi-agent frameworks further decompose planning: Process of Multi-agent Collaboration (PMC; (Dang et al., 2025)) performs LLM-based task delegation but relies on emergent decomposition, while

*Table 1.* Feature-level comparison of representative agentic planning methods. ✓ denotes the method supports the feature. CM = Constraint Manager.

| Method | Hierarchical (Coord/Work) | Parallel Execution | Context Isolation | Global Verifier/CM | Iterative Refine/Neg. | RL Trained |
|---|---|---|---|---|---|---|
| ReAct (Yao et al., 2022) | | | | | | |
| Reflexion (Shinn et al., 2023) | | | | | ✓ | |
| PMC (Dang et al., 2025) | ✓ | | | | | |
| EvoAgent (Yuan et al., 2025) | | | | | ✓ | |
| HIPLAN (Li et al., 2025) | ✓ | | | | ✓ | |
| MTP (Zhang et al., 2024) | ✓ | | | | ✓ | |
| ATLAS (Choi et al., 2025) | ✓ | | | ✓ | ✓ | |
| DeepTravel (Ning et al., 2025) | | | | ✓ | ✓ | ✓ |
| **HiMAP-Travel (ours)** | ✓ | ✓ | ✓ | ✓ | ✓ | ✓ |

MTP formalizes Manager/Executor/Supervisor-style roles but typically uses static sub-plan assignment and natural language message passing. Such hierarchical multi-agent approaches remain brittle under *dynamic coupled constraints* (e.g., shared budgets or global non-duplication), where coordination via verbal negotiation is slow and error-prone. HiMAP-Travel addresses this with a **Synchronized Global State** for deterministic invariant enforcement and a **Cooperative Bargaining Protocol** for structured re-allocation, achieving 52.78% validation FPR and 52.65% test FPR with near-constant latency scaling via parallel execution.

Table 1 provides a feature-level comparison of representative methods and situates HiMAP-Travel's design choices.

## 3. Preliminaries and Problem Formulation

We formalize long-horizon autonomous planning as a **Hierarchical Constrained Goal-Conditioned Partially Observable Markov Decision Process (HC-POMDP)**. Unlike standard reinforcement learning tasks with scalar reward maximization, travel planning requires synthesizing a semantic trajectory $\tau$ that satisfies rigid logical invariants (hard constraints) while optimizing user utility (soft constraints). Let $\mathcal{W}$ denote the latent world state, only partially observable through API interfaces $\mathcal{K}$. The agent receives a natural language query $q$, which parameterizes $\mathcal{T}_q = \langle q, \mathcal{K}, \mathcal{C}_{hard}, \mathcal{C}_{soft} \rangle$. The hard constraint function $\mathcal{C}_{hard} : \Omega \rightarrow \{0, 1\}$ filters inviolable logic (e.g., budget caps, temporal causality), while $\mathcal{C}_{soft} : \Omega \rightarrow \mathbb{R}$ captures preferences. The objective is to find $\pi^*$ that maximizes expected utility subject to near-sure validity: $\pi^* = \arg\max_\pi \mathbb{E}_{\tau \sim \pi}[\mathcal{C}_{soft}(\tau)]$, subject to $P(\mathcal{C}_{hard}(\tau) = 1) \geq 1 - \delta$.

The central challenge for monolithic LLM planners is **Constraint Drift under Long Tool Traces**. In sequential planners (e.g., ReAct), the policy $\pi_\theta(a_t | h_{t-1})$ conditions on the full history $h_{t-1}$. As the horizon $T$ extends, intermediate tool outputs and reasoning traces accumulate, increasing the effective context length. We posit that attention to the initial global constraints (e.g., the total budget specified in $q$) diminishes as the sequence grows, causing the model to prioritize local coherence over global feasibility. Consequently, the probability of generating a valid trajectory decreases with horizon length, making monolithic planning brittle for multi-day itineraries.

To assess performance, we adopt standard metrics from the TravelPlanner benchmark. The primary metric is **Final Pass Rate (FPR)**, the percentage of queries for which the agent generates a complete itinerary satisfying all hard constraints (budget, duration, required cities, etc.). We also report **Delivery Rate**, the percentage of queries with a syntactically valid final answer regardless of constraint satisfaction. For constraint-specific analysis, we track **Valid Route** (geographic/temporal feasibility) and **Budget Adherence** (total cost within the specified budget).

## 4. Methodology: HiMAP-Travel

To address the computational intractability of monolithic long-horizon travel planning, we propose **HiMAP-Travel** (Figure 1), a hierarchical architecture that decouples planning into two functional levels: a *Strategic Level* (resource allocation) and a *Tactical Level* (atomic execution). This separation enables the parallel resolution of temporal sub-problems, effectively transforming the horizon from $T$ to $T/D$, where $D$ is the number of travel days. Parallelization, however, introduces coupled global constraints (shared budget, non-duplication, mode consistency) and a sparse, delayed reward that depends on *joint* satisfaction. We resolve these challenges with (i) a lightweight synchronized monitor $\Sigma$ that deterministically prevents resource conflicts, (ii) an iterative cooperative bargaining protocol for feasibility-driven re-allocation, and (iii) a unified, parameter-shared transformer policy trained via Group Relative Policy Optimization (GRPO; (Shao et al., 2024)), with a memory-

efficient multi-role update mechanism.

## 4.1. Hierarchical Agents: Coordinator and Parallel Executors

**Coordinator (Strategic Planning).** Given user query $q$, the Coordinator projects $q$ into a structured latent plan that partitions global requirements into day-level boundary conditions. Let the query impose a global budget $B_{total}$ and a set of required destinations. The Coordinator synthesizes a meta-plan $\mathcal{Z} = \{z_1, \ldots, z_D\}$, where each $z_d$ specifies (i) the target city, (ii) the day's semantic role (e.g., *Departure*, *Full-Stay*, *Transit*), and (iii) a *budget hint* $b_d$ satisfying a conservation law $\sum_d b_d \leq B_{total}$. Intuitively, the Coordinator "pre-solves" the hardest global constraints at the latent level by enforcing budget feasibility and coarse route structure before any tactical actions are taken. In practice, $b_d$ serves as a *soft prior* that discourages expensive choices early; it never invalidates an otherwise feasible action. Hard coupled constraints are enforced only by the synchronized monitor through atomic commit checks (Section 4.3). This separation is important: the Coordinator shapes the search distribution, while $\Sigma$ provides the runtime safety gate.

**Executors (Parallel Tactical Resolution).** The execution layer consists of $D$ Day Planners (Executors). Each Executor conditions on sub-goal $z_d$ and generates a day-level trajectory $\tau_d$ within a strictly isolated MDP. In a monolithic planner, the state at time $t$ accumulates all prior days' reasoning traces, leading to context saturation. HiMAP-Travel enforces *Context Independence*: action generation on Day $d$ is independent of other days' internal reasoning traces, preventing "context rot" where early hallucinations contaminate later decisions. Cross-day coupling is handled *only* through the shared monitor $\Sigma$ and the bargaining protocol, not via verbose inter-agent message passing.

## 4.2. Single-Policy Role Conditioning and Concurrency

Rather than maintaining separate parameter sets for the Coordinator and Executors, we unify them into a single policy $\pi_\theta$ and elicit distinct roles through *System Prompt Conditioning*. The policy input is $x = [p_{role}, s_t]$, where $p_{role}$ describes the role scope (Coordinator vs. Day Planner) and $s_t$ is the current observation/state. This design enables transfer: reasoning learned during tactical execution (e.g., recognizing expensive flights) can inform strategic allocation under the Coordinator role prompt.

During inference, we instantiate $D+1$ logical threads of $\pi_\theta$ (one Coordinator + $D$ Executors). To control compute, we cap executor concurrency to $P = 3$ workers and schedule $D$ day planners in $\lceil D/P \rceil$ batches.

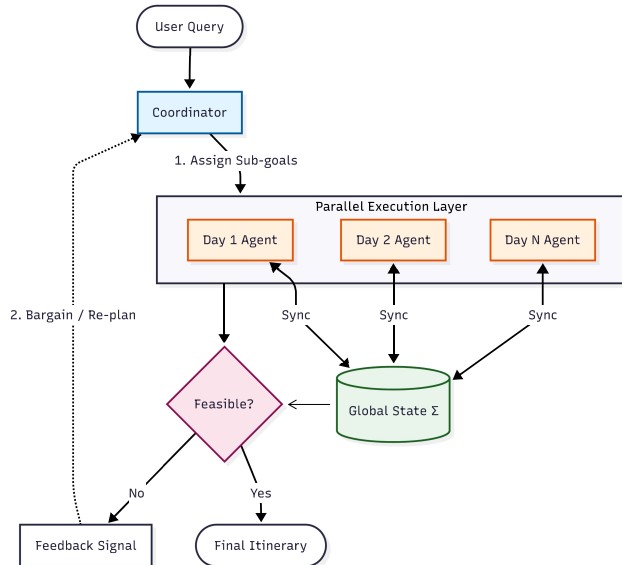

*Figure 1.* The HiMAP-Travel Architecture. The Coordinator projects the query into sub-goals $z_d$. Parallel Executors generate day-plans $\tau_d$ while synchronizing via the Global State $\Sigma$. If constraints are violated, the Bargaining Protocol triggers re-allocation.

## 4.3. Synchronized Global State Mechanism

Parallel execution under shared resources faces a "tragedy of the commons" (Hardin, 1968): independent day planners may overspend or double-book. HiMAP-Travel introduces a **Synchronized Global State** $\Sigma$ that acts as a deterministic transactional monitor. Unlike latent neural memory or natural-language blackboards, $\Sigma$ is an external structured store protected by a global re-entrant mutex.

**Definition (Constraint Monitor Contract).** Let $\Sigma = \langle B_{used}, V_{committed}, M_{trans} \rangle$ track cumulative budget, the set of globally committed venues, and an inter-city transport-mode lock. Table 2 summarizes the full interface. The monitor enforces invariants $I(\Sigma)$ (budget conservation, non-duplication, mode consistency) via atomic operations: The CHECK($a$) operation validates an action without mutation and returns success or a typed error; the COMMIT($a$) operation atomically updates $\Sigma$ iff $I(\Sigma')$ holds, otherwise rejecting with an error code. For iterative re-planning, we additionally support CHECKPOINT() and ROLLBACK() operations. Rollback assumes tentative actions can be released cost-free, consistent with common reserve-then-confirm workflows.

**Runtime interception.** When an Executor samples an action $a_t$ (e.g., finishing with a specific restaurant), the environment intercepts $a_t$, acquires the global lock, and validates $a_t$ against $\Sigma$. If the venue already exists in $V_{committed}$ or the budget would overflow, the action is rejected with a structured "Constraint Violation" observation, prompting regeneration. This yields deterministic conflict prevention

*Table 2.* Synchronized Global State interface used by $\Sigma$.

| State Fields | Description |
|---|---|
| $B_{total}, B_{used}$ | Global budget cap and cumulative spend |
| $V_{committed}$ | Set of committed restaurants and attractions |
| $M_{trans}$ | Inter-city transport-mode lock |

| Operations | Semantics |
|---|---|
| check(action) | Validate action against $I(\Sigma)$; return OK or typed error |
| commit(action) | Atomically update $\Sigma$ if valid; otherwise reject with error code |
| checkpoint() | Save a rollback point for iterative re-planning |
| rollback() | Restore latest checkpoint if bargaining fails |

---

**Algorithm 1** HiMAP-Travel with Cooperative Bargaining

1: **Initialize** $\Sigma \leftarrow \{B_{used} = 0, V_{committed} = \emptyset, M_{trans} = \emptyset\}$
2: **Input** Query $q$, Max Iterations $K_{max}$
3: **for** $k = 1$ **to** $K_{max}$ **do**
4:    $\mathcal{Z}^{(k)} \leftarrow \text{Coordinator}(q, \text{Feedback}^{(k-1)})$
5:    $\Sigma.\text{checkpoint}()$
6:    **Parallel For** $d = 1 \dots D$:
7:      $\tau_d, \text{status}_d \leftarrow \text{Executor}_d(z_d^{(k)}, \Sigma)$
8:      **if** $\text{status}_d ==$ INFEASIBLE **then**
9:        CancelOtherThreads(); $\Sigma.\text{rollback}()$; **break**
10:    **if** $\forall d, \text{status}_d ==$ FEASIBLE **then**
11:      **Return** $\bigcup_d \tau_d$
12:    **else**
13:      $\text{Feedback}^{(k)} \leftarrow \bigcup_d \text{ExtractFeedback}(\tau_d)$
14:      $\Sigma.\text{rollback}()$
15:    **end if**
16: **end for**
17: **Return** $\arg\max_{\tau \in \{\tau^{(1)}\dots\tau^{(K)}\}} \text{Reward}(\tau)$

---

without requiring inter-agent negotiation for every action.

**Safety guarantee.** Assuming all mutations to $\Sigma$ are serialized through the global mutex and COMMIT validates $I(\Sigma')$ before mutation, no two Executors can successfully commit conflicting monitor-tracked actions, and the final monitor state preserves the budget, non-duplication, and mode-consistency invariants regardless of thread interleaving.

**Scope.** Critically, $\Sigma$ prevents only *resource conflicts* (budget exhaustion, double-booking, mode inconsistency) and does not solve travel planning. Route feasibility, temporal consistency, cuisine coverage, and other commonsense constraints remain the responsibility of the learned policy and are checked by the environment evaluator. Formal safety guarantees, canonicalization rules, and sensitivity analysis are provided in Appendix A.

### 4.4. Cooperative Bargaining Protocol

A strict top-down hierarchy can be brittle when the Coordinator assigns infeasible sub-goals. We therefore introduce a **Cooperative Bargaining Mechanism** that enables bidirectional feasibility feedback: Executors can reject infeasible assignments using lightweight, typed violation signals, and the Coordinator revises the task structure (e.g., selecting different cities/routes or adjusting day roles) for the next iteration.

Executors return feedback using a strict JSON schema with fields: *status* ("feasible" or "infeasible"), *deficit* (estimated budget shortfall), and *violation_type* ("budget", "time", or "availability"). Communication is restricted to these lightweight payloads rather than verbose natural language, minimizing token overhead while preserving parallel execution.

### 4.5. Hierarchical Reward Decomposition

To address credit assignment in hierarchical RL, we decompose rewards into global and local signals. The Coordinator optimizes $J_{coord} = \lambda_{global}R_{global} + \lambda_{extract}R_{extract} - \lambda_{iter}N_{iter}$, where $R_{extract}$ rewards constraint extraction and $N_{iter}$ is the number of bargaining iterations (penalizing excessive re-planning encourages high-quality initial allocations). Executors optimize $J_{exec} = \gamma_{global}R_{global} + \gamma_{local}R_{local} + \gamma_{early}\mathbb{K}_{early}$, where $\mathbb{K}_{early}$ rewards fail-fast detection. The global reward $R_{global} = \mathbb{I}(\text{Valid}) + \alpha \sum_c s(c) - \beta \max(0, B_{used} - B_{total})$ combines constraint satisfaction with soft preferences. The local executor reward is a day-level dense signal:

$$R_{local}(\tau_d) = \sum_{c \in \mathcal{C}_{local}(d)} w_c s_c(\tau_d) - \eta \max(0, \hat{C}_d - b_d),$$

where $\mathcal{C}_{local}(d)$ includes day-local cuisine, room type, house-rule, within-city, and intra-day consistency constraints; $s_c(\tau_d) \in [0, 1]$ is the satisfaction score; $\hat{C}_d$ is the cost returned by cost_enquiry; and $b_d$ is the Coordinator's soft budget hint. Thus $R_{local}$ encourages locally useful plans without overriding the hard global cap enforced by $\Sigma$. Full reward hyperparameters are provided in Appendix A.2.

### 4.6. Group Relative Policy Optimization (GRPO)

We train $\pi_\theta$ using GRPO, eliminating the need for a separate critic. For each query $q$, we sample a group of $G$ trajectories $\{\tau_i\}_{i=1}^G$ and compute group-relative advantages $A_i = (R(\tau_i) - \mu_G)/(\sigma_G + \epsilon)$, where $\mu_G$ and $\sigma_G$ are the group mean and standard deviation of rewards. This group baseline acts as a dynamic curriculum: even when all trajec-

tories fail early in training, the least-bad trajectory receives positive advantage. We apply a KL penalty with coefficient $\beta$ to prevent drift from the base model; hyperparameters are listed in Appendix A.

### 4.7. Memory-Efficient Multi-Agent Policy Updates

A critical challenge is memory usage when updating $D+1$ logical agents under a shared policy $\pi_\theta$. A naive implementation that accumulates all $(D+1) \times G$ trajectories before each update requires storing approximately $(D+1) \times G \times T \times d_{model}$ activations, where $T$ is the average trajectory length and $d_{model}$ is the hidden dimension, which can exceed single-node memory budgets for long trajectories.

We introduce a **Shared Rollout Buffer** with a **First-In-First-Update (FIFO)** mechanism. During each iteration, trajectories from all roles are appended to a unified buffer $\mathcal{B}$ and partitioned by role prompt. We maintain role-specific group counters $\{n_{coord}, n_{exec}^{(1)}, \ldots, n_{exec}^{(D)}\}$. As soon as any role partition accumulates $G$ trajectories, we immediately compute GRPO group statistics *within that partition*, perform a gradient update on $\pi_\theta$ conditioned on the corresponding role prompt, and *flush* those trajectories. This bounds peak memory by $\max(G \times T) \times d_{model}$ per role rather than the full cross-product, reducing peak memory by a factor of $(D+1)$.

FIFO updates also handle heterogeneous completion times: Executors with simpler $z_d$ may finish earlier than the Coordinator. Updating each role as soon as its group is ready maximizes throughput without stalling on the slowest role, while preserving GRPO validity because each group consists of comparable trajectories generated under $\pi_\theta(\cdot \mid p_{role})$.

### 4.8. Training Details

We train on the TravelPlanner training split (45 queries) and evaluate Qwen3-4B-Instruct-2507 and Qwen3-8B (Yang et al., 2025). We fine-tune for 100 epochs with Adam (Kingma, 2014) (learning rate $5 \times 10^{-6}$, batch size 32, gradient accumulation steps 4) under an 8192-token context window. For GRPO we use group size $G = 4$ and KL coefficient $\beta = 0.01$. Decoding uses temperature $T = 0.7$, top-p $p = 0.9$. We cap executor tool calls at 15 per day-agent and bargaining iterations at $K_{max} = 3$. Additional implementation notes, overfitting analysis, and system configuration are provided in Appendix A (see also Appendix B and Appendix B.4).

## 5. Empirical Evaluation

**Benchmarks.** We evaluate on two complementary benchmarks: TravelPlanner (Xie et al., 2024) (1,225 queries: 45

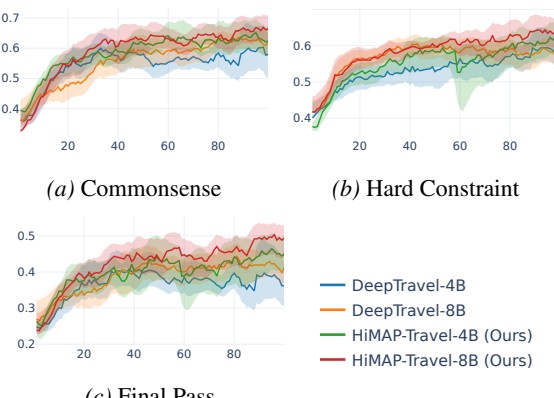

*(a)* Commonsense      *(b)* Hard Constraint

*(c)* Final Pass

*Figure 2.* **Training Dynamics.** HiMAP-Travel (Red/Green) achieves superior Hard Constraint convergence vs sequential baseline.

train, 180 val, 1000 test) requires single-turn planning under 13 coupled constraints (8 commonsense, 5 hard including budget, diversity, room rules); FlexTravelBench (Oh et al., 2025) tests multi-turn constraint adaptation across 4 scenarios where constraints are progressively revealed. Success requires satisfying all constraints simultaneously; budget compliance accounts for 45% of failures.

**Baselines & Evaluation Protocol.** We report results for: (1) proprietary and prior-work baselines (e.g., GPT-4-Turbo (OpenAI, 2023), Gemini-2.5-Pro (Google DeepMind, 2025), ReAct/Reflexion/ATLAS/MTP) as published in their respective papers under the TravelPlanner/FlexTravelBench metrics; and (2) a **controlled comparison** between HiMAP-Travel and a sequential RL baseline based on DeepTravel, where we hold constant the backbone model (Qwen3-4B/8B), training algorithm (GRPO), training split (45 queries), tool APIs, and decoding.

**Fairness and Information Access.** Both HiMAP-Travel and the controlled DeepTravel baseline interact with the same external environment tools (flight/hotel/restaurant search and cost enquiry) and are evaluated by the official harness on the same database version. The Synchronized Global State $\Sigma$ is *internal* to HiMAP-Travel: it only records (i) cumulative cost derived from agent-proposed actions, (ii) canonicalized venue identifiers from previously committed actions, and (iii) a transport-mode lock. It does not query any additional tools, access ground-truth labels, or inspect database contents beyond what the agent already retrieved. Its only effect is to accept/reject an agent-proposed commit operation and return a typed violation observation (e.g., budget overflow, duplicate venue), enabling early correction during parallel execution.

**Implementation (Controlled Runs).** For HiMAP-Travel and the controlled DeepTravel baseline, we use TravelPlanner v1.0 with the official evaluation harness, identical de-

*Table 3.* Performance on TravelPlanner Benchmark. We report results on both validation and test splits for representative methods. DeepTravel and HiMAP-Travel results are evaluated under the official TravelPlanner harness with identical experimental conditions. Complete baseline comparison is provided in Appendix C.2.

| Method | Delivery | Validation (180 queries) | | | | | Test (1000 queries) | | | | |
|---|---|---|---|---|---|---|---|---|---|---|---|
| | | Commonsense | | Hard Constr. | | FPR | Commonsense | | Hard Constr. | | FPR |
| | | Micro | Macro | Micro | Macro | | Micro | Macro | Micro | Macro | |
| *Multi-Agent Frameworks (Published Results)* | | | | | | | | | | | |
| MTP (GPT-4 w/ hint) | 96.7/97.4 | 87.29 | 43.33 | 47.14 | 46.67 | 33.33 | 91.46 | 50.00 | 53.96 | 45.12 | 42.68 |
| ATLAS (Gemini-2.5-Pro) | 100/100 | 48.33 | 88.54 | 82.62 | 74.44 | 44.44 | 85.81 | 40.50 | 77.64 | 70.60 | 35.00 |
| *Sequential RL Baseline (Our Evaluation)* | | | | | | | | | | | |
| DeepTravel (Qwen3-4B) | 96.8/98.2 | 90.00 | 59.58 | 63.15 | 49.44 | 35.28 | 90.62 | 53.85 | 39.50 | 52.33 | 34.92 |
| DeepTravel (Qwen3-8B) | 97.6/99.4 | 92.60 | 67.64 | 71.43 | 58.61 | 45.56 | 93.28 | 60.62 | 45.66 | 60.12 | 43.98 |
| *HiMAP-Travel (Ours)* | | | | | | | | | | | |
| HiMAP-Travel (Qwen3-4B) | 100/100 | 95.57 | 72.36 | 70.89 | 59.17 | 48.61 | 94.72 | 66.03 | 46.44 | 63.45 | 49.80 |
| **HiMAP-Travel (Qwen3-8B)** | **100/100** | **95.64** | **73.61** | **76.85** | **63.33** | **52.78** | **94.62** | **67.00** | **50.47** | **66.05** | **52.65** |

coding (temperature 0.7, top-p 0.9) and tool-call budgets (max 15 tool calls per day; sequential baseline receives the same total tool-call budget matched to trip duration). We report mean ± std across 4 seeds.

## 5.1. Comparative Performance Analysis (RQ1)

Table 3 presents our main results on TravelPlanner, comparing HiMAP-Travel against representative state-of-the-art methods on both validation (180 queries) and test (1000 queries) splits. HiMAP-Travel achieves **52.78% validation FPR** and **52.65% test FPR** with Qwen3-8B, demonstrating strong generalization with only 0.13 pp gap. Complete results including all baselines are provided in Appendix C.2.

Relative to published results, the method is higher than ATLAS by +17.65 pp and MTP by +10.0 pp; the controlled architectural comparison is the DeepTravel baseline, where HiMAP-Travel gains +8.67 pp (52.65% vs 43.98%) with substantially lower variance across 4 seeds (std: 0.48% vs 7.18%) and perfect 100% Delivery Rate. The 8B vs 4B capacity gap (+2.85 pp overall, +4.03 pp on Hard Constraints Micro) reveals strategic coordination benefits from increased model capacity. Training dynamics (Figure 2) validate this architectural advantage: while both converge on Commonsense Constraints (~65%), HiMAP-Travel sustains ~70% Hard Constraint satisfaction versus DeepTravel's ~60%, confirming hierarchical decomposition stabilizes coupled constraint handling by decoupling the planning horizon into manageable sub-problems.

**Baseline Analysis.** Against published results, HiMAP-Travel surpasses MTP (which requires manual prompt engineering) and ATLAS (which incurs high latency). The controlled DeepTravel comparison—identical backbone, GRPO, and tools—provides the primary architectural evidence: HiMAP-Travel gains +8.67 pp (52.65% vs. 43.98%) with 93% variance reduction (std: 0.48% vs. 7.18%). The

4B gap is even larger (+14.88 pp), confirming smaller models benefit more from hierarchical factorization. ATLAS and MTP comparisons reflect both architectural and model-capacity differences and are included as reference points only.

**Constraint-Level and Difficulty Insights.** Beyond aggregate FPR, the gains concentrate on constraints that require global coordination. On the validation split, Qwen3-8B improves over 4B on Budget (+5.62 pp), Cuisine (+6.59 pp), Complete Information (+4.42 pp), and Transportation (+2.94 pp), while diversity and route constraints are already near saturation (> 98%). The remaining bottlenecks are fine-grained accommodation fields: Room Type remains below 43%, and Room Rule decreases from 37.92% to 32.73%, suggesting that sparse database-field constraints are harder to learn from the 45-query training split than global resource constraints. This also explains why Hard Constraint Micro is more sensitive to split composition than FPR.

Difficulty-wise, HiMAP-Travel improves over DeepTravel across all test tiers: +11.1 pp on Easy, +11.5 pp on Medium, and +3.4 pp on Hard queries. The smaller Hard-tier gain reflects genuine combinatorial difficulty, but the variance reduction remains substantial: on Hard queries, DeepTravel's FPR standard deviation is 12.3%, compared with 1.5% for HiMAP-Travel. Thus the hierarchy's main empirical effect is not only higher mean success, but also more stable constraint satisfaction under increasing coupling.

## 5.2. FlexTravelBench: Multi-Turn Adaptation

Table 4 shows HiMAP-Travel achieves **44.34%** (2-turn, +5.14 pp vs ATLAS) and **37.42%** (3-turn, +4.72 pp) with consistent gains across all constraint adaptation scenarios: Local Add (44.96%), Global Add (43.72%), Local→Global (37.42%), and Global→Local (37.32%).

*Table 4.* Performance on FlexTravelBench: Multi-Turn Constraint Adaptation. Results show performance across 2-turn (single constraint addition) and 3-turn (sequential constraint revelation) scenarios. Numbers in parentheses indicate instance count. ATLAS and ReAct report per-constraint results (Cuisine, Room rule, Room type for Local; Budget, Number of People for Global). HiMAP-Travel reports aggregated results across all constraint types within each scenario. Micro: weighted by instance count; Macro: averaged across seeds.

| Scenario | Method | Delivery | Commonsense | | Hard Constraint | | Final Pass |
| | | | Micro | Macro | Micro | Macro | |
|---|---|---|---|---|---|---|---|
| *2-Turn Local: Add local constraint (Cuisine, Room rule, or Room type) in Turn 2* | | | | | | | |
| Local Add Avg (#189) | ReAct (GPT-4) | 100.00 | 85.84 | 44.41 | 67.29 | 43.72 | 29.58 |
| | ATLAS (Gemini-2.5-Pro) | 100.00 | 75.48 | 59.80 | 77.23 | 64.62 | 38.86 |
| | HiMAP-Travel (4B) | 100.00 | 76.80 | 61.25 | 79.15 | 66.50 | 41.25 |
| | **HiMAP-Travel (8B)** | **100.00** | **78.62** | **62.15** | **80.35** | **67.40** | **44.96** |
| *2-Turn Global: Add global constraint (Budget or Number of People) in Turn 2* | | | | | | | |
| Global Add Avg (#240) | ReAct (GPT-4) | 100.00 | 61.93 | 63.75 | 51.67 | 68.34 | 26.25 |
| | ATLAS (Gemini-2.5-Pro) | 100.00 | 67.09 | 69.22 | 70.14 | 70.00 | 39.58 |
| | HiMAP-Travel (4B) | 100.00 | 68.20 | 70.15 | 71.45 | 71.20 | 40.85 |
| | **HiMAP-Travel (8B)** | **100.00** | **69.45** | **71.30** | **72.80** | **72.35** | **43.72** |
| *3-Turn: Local constraint (Turn 2), then Global constraint (Turn 3)* | | | | | | | |
| Local→Global (#378) | ReAct (GPT-4) | 100.00 | 83.70 | 33.07 | 59.59 | 36.51 | 15.34 |
| | ATLAS (Gemini-2.5-Pro) | 100.00 | 87.96 | 49.21 | 73.58 | 53.97 | 33.60 |
| | HiMAP-Travel (4B) | 100.00 | 88.65 | 50.35 | 74.80 | 55.15 | 35.20 |
| | **HiMAP-Travel (8B)** | **100.00** | **89.35** | **51.60** | **75.80** | **56.25** | **37.42** |
| *3-Turn: Global constraint (Turn 2), then Local constraint (Turn 3)* | | | | | | | |
| Global→Local (#378) | ReAct (GPT-4) | 100.00 | 84.06 | 32.80 | 59.43 | 36.24 | 17.20 |
| | ATLAS (Gemini-2.5-Pro) | 100.00 | 86.81 | 47.09 | 71.38 | 52.12 | 31.75 |
| | HiMAP-Travel (4B) | 100.00 | 87.50 | 48.20 | 72.60 | 53.35 | 33.15 |
| | **HiMAP-Travel (8B)** | **100.00** | **88.20** | **49.80** | **73.65** | **54.40** | **37.32** |

The 3-turn performance validates the Cooperative Bargaining Protocol's ability to trigger checkpoint/rollback when new constraints invalidate existing plans, while the 4B model achieving 41.05% (2-turn) and 34.18% (3-turn) confirms hierarchical benefits scale to smaller backbones. The validation-test consistency (52.78% vs 52.65%, 0.13 pp gap) with low cross-seed variance (0.79% val, 0.48% test std) demonstrates robust generalization.

## 5.3. Failure Analysis

Sequential baselines exhibit constraint drift with budget satisfaction degrading from 98% (Day 1) to 42% (Day 5) as context saturates, whereas HiMAP-Travel maintains $> 90\%$ via isolated contexts. Manual analysis of 200 failed instances shows that the main residual errors are budget overflow, duplicate venues, and route/temporal inconsistencies (Table 5). The monitor directly targets two dominant categories: budget-overflow rates fall from 12.5% to 4.1%, and duplicate-venue rates fall from 8.7% to 1.5%. Minimum-night violations remain a limitation because they require lookahead over the Coordinator's city-day schedule rather than only resource-conflict tracking.

## 5.4. Efficiency (RQ2)

HiMAP-Travel achieves **2.63×** wall-clock speedup on 7-day trips (72s vs 189.5s DeepTravel); in our setup executor

*Table 5.* Failure taxonomy from manual analysis of 200 failed test instances, with comparative rates against DeepTravel.

| Failure Category | % Failures | DeepTravel Rate | HiMAP Rate |
|---|---|---|---|
| Budget overflow | 34.0% | 12.5% | **4.1%** |
| Early overspend | 21.0% | 8.3% | **1.8%** |
| Duplicate venues | 23.5% | 8.7% | **1.5%** |
| Route infeasibility | 14.5% | 3.8% | **1.1%** |
| Missing required cities | 11.5% | 2.9% | **0.8%** |
| Min. nights violations | 10.5% | 2.6% | **0.9%** |
| Constraint conflicts | 6.0% | 1.5% | **0.4%** |

concurrency is capped at $P = 3$, so the day-planning phase has a theoretical upper bound of 3× speedup. We observe sub-linear scaling (3/5/7-day: 48/58/72s) versus sequential linear $O(D)$ growth (98/142/190s). Bargaining overhead remains modest at 10.8% of wall-clock time, with 61.7% of validation queries resolved without feedback and 97.8% resolved by $K_{max} = 3$ (Table 6). Table 7 demonstrates 2.5-3.8× latency reduction at moderate compute cost, while the 4B model delivers 88% of 8B performance with 67% latency and 57% compute requirements. Complete phase-by-phase latency breakdown and scaling analysis provided in Appendix C.1.

## 5.5. Ablations (RQ3)

Table 8 quantifies individual component contributions: removing Global State causes -9.58 pp degradation (diversity violations: 8→34%); removing Coordinator yields -

*Table 6.* Bargaining convergence on validation queries.

| Stage | Cumulative Resolved | Marginal |
|---|---|---|
| Initial allocation | 111/180 (61.7%) | 111 |
| After feedback 1 | 162/180 (90.0%) | 51 |
| After feedback 2 | 176/180 (97.8%) | 14 |
| Unresolved at cap | 4/180 (2.2%) | – |

*Table 7.* Computational Efficiency Metrics (7-Day Itinerary)

| Method | Latency (s) | Compute (s) | Tokens (k) |
|---|---|---|---|
| ReAct (Qwen3-8B) | 180 | 180 | 12.5 |
| DeepTravel (4B) | 165 | 165 | 11.8 |
| DeepTravel (8B) | 190 | 190 | 12.3 |
| **HiMAP-Travel (4B)** | **48** | 120 | 13.8 |
| **HiMAP-Travel (8B)** | **72** | 210 | 14.2 |

12.98 pp (budget failures); disabling bargaining reduces performance by -3.88 pp; eliminating parallelism costs -7.18 pp. These controlled ablations validate each architectural choice; comprehensive component-wise analysis with detailed failure modes provided in Appendix D.

*Table 8.* Ablation Study (Qwen3-8B, Validation)

| Configuration | FPR | Δ |
|---|---|---|
| **Full System** | **52.78%** | — |
| w/o Monitor | 43.2% | -9.58 pp |
| w/o Coordinator | 39.8% | -12.98 pp |
| w/o Bargaining | 48.9% | -3.88 pp |
| w/o Parallelism | 45.6% | -7.18 pp |

The Coordinator and monitor fail in different ways, indicating complementary roles rather than redundant constraint handling. Removing the Coordinator primarily causes allocation errors: budget violations increase to 28.4% and cascading failures to 23.4%, because early days overspend before downstream needs are known. Removing $\Sigma$ primarily causes parallel execution conflicts: restaurant duplication increases from 7.7% to 34.1% and attraction duplication from 9.9% to 28.7%, even when the Coordinator provides reasonable sub-goals. Thus $b_d$ improves strategic allocation, whereas $\Sigma$ prevents race conditions that individually reasonable Executors cannot avoid without shared atomic state.

## 6. Discussion and Broader Impact

HiMAP-Travel demonstrates that hierarchical decomposition with transactional constraint monitoring enables scalable planning under coupled constraints. Controlled comparison isolates the architectural contribution: using identical backbones and training procedures, the hierarchical approach outperforms sequential planning by 8.67 pp with 93% variance reduction across seeds.

**Key Insights.** The Synchronized Global State trades seman-tic flexibility for deterministic safety, preventing resource conflicts through atomic operations. Removing it causes 9.58 pp degradation while maintaining 43.2% FPR, confirming the policy handles most planning complexity autonomously. The Cooperative Bargaining Protocol unifies single-turn constraint satisfaction and multi-turn adaptation under iterative refinement. Single-policy role conditioning yields an 8.08 pp gain over separately trained role policies in our ablation, although the precise mechanism behind this benefit remains an open question.

**Generalization.** Our empirical evaluation is limited to travel planning and multi-turn travel adaptation. The method is expected to transfer to tasks with decomposable sub-problems, coupled resources, and parallel conflict risk–for example, software module planning, supply-chain routing, or experiment scheduling–but validating these domains requires separate tool libraries and evaluation harnesses. We therefore treat cross-domain evaluation as future work, while using FlexTravelBench to test a qualitatively different multi-turn adaptation setting within the travel domain.

**Limitations.** HiMAP-Travel is designed for tasks whose global constraints can be represented as shared resources and whose horizon admits meaningful decomposition. The current monitor intentionally covers budget, venue non-duplication, and transport-mode consistency; temporal constraints that require lookahead over the city-day schedule, such as accommodation minimum-night rules, remain handled by the learned policy and external evaluator. Training uses the official 45-query TravelPlanner split with rollout augmentation, which is sufficient for controlled comparison but leaves rare constraints relatively sparse. Finally, parallel execution reduces wall-clock latency while modestly increasing total token and compute usage, reflecting a throughput trade-off rather than a free efficiency gain.

## 7. Conclusion

This work targets a key weakness of LLM-based planners on long-horizon tasks: *Constraint Drift under Long Tool Traces*, where adherence to global hard constraints degrades as intermediate outputs accumulate in context. We propose **HiMAP-Travel**, a hierarchical multi-agent framework that decouples strategic coordination from parallel day-level execution, enabled by a **Synchronized Global State** for deterministic enforcement of coupled invariants, a **Cooperative Bargaining Protocol** for structured re-allocation when sub-goals are infeasible, and a **Unified Role-Conditioned Policy** trained with GRPO. HiMAP-Travel achieves 52.65% test Final Pass Rate on `TravelPlanner` and strong `FlexTravelBench` results while reducing latency by 2.5×, demonstrating that long-horizon reliability benefits from architectural factorization and deterministic constraint enforcement rather than post-hoc repair.

## Impact Statement

This paper advances hierarchical multi-agent planning for long-horizon constrained tasks. While our evaluation focuses on travel planning as a rigorous testbed, the methodology may be adapted to domains requiring resource coordination under coupled constraints, such as project management, supply chain optimization, and scientific experiment planning. The societal implications are consistent with those of AI planning systems generally. We do not identify specific negative impacts warranting additional discussion beyond standard considerations for autonomous decision-making systems.

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

# Appendix Overview

This appendix provides comprehensive technical documentation organized into six major parts. Implementation details are presented in Section A, including the Synchronized Global State $\Sigma$ specification, reward decomposition, training hyperparameters, memory-efficient multi-agent updates, and the complete tool library specification. Dataset and benchmarks are documented in Section B, covering task setups, constraint structures, full evaluation protocol with constraint taxonomy, dataset statistics, and training data construction for both TravelPlanner and FlexTravelBench. Section C offers extensive experimental results with complete baseline comparisons, constraint-level performance analysis, and efficiency measurements. Comprehensive component-wise ablations with statistical analysis are detailed in Section D. System prompts and execution traces are presented in Section E, including role-conditioned coordinator and executor prompts, step-by-step tool-call traces, and cooperative bargaining protocol examples. Finally, systematic failure taxonomy with quantitative case studies and current limitations are presented in Section F.

# A. Implementation Details

This section provides complete technical specifications enabling full reproducibility of our results. We detail the Synchronized Global State $\Sigma$ mechanism with formal safety guarantees, hierarchical reward decomposition strategy, training hyperparameters and optimization details, memory-efficient FIFO update mechanism for multi-agent policy training, and the complete tool library specification. These implementation details are critical for understanding the system's behavior and replicating our experimental results.

## A.1. Synchronized Global State Mechanism

HiMAP-Travel introduces the **Synchronized Global State** $\Sigma$ as a deterministic transactional monitor to prevent resource conflicts under parallel execution. Unlike latent neural memory or natural language blackboards, $\Sigma$ is an external structured data store that enforces hard constraints via atomic operations protected by a global re-entrant mutex.

The monitor tracks three critical global variables. First, a **Diversity Set** contains globally committed restaurant and attraction names to prevent duplicate bookings. Second, a **Budget Ledger** records cumulative cost across all agents. Third, a **Transport Mode Lock** ensures consistency across days. The global state is formally defined as $\Sigma = \langle B_{used}, V_{committed}, M_{trans} \rangle$, where $B_{used} \in \mathbb{R}_{\geq 0}$ tracks cumulative budget expenditure across all agents, $V_{committed} \subset \mathcal{V}$ maintains the set of globally committed venues to enforce diversity constraints, and $M_{trans}$ stores the inter-city transportation mode lock to ensure consistency.

The full interface is summarized in Table 2; this appendix expands on its implementation semantics, validation rules, and safety guarantees.

The monitor enforces global invariants $I(\Sigma)$ including budget constraints ($B_{used} \leq B_{total}$) and non-duplication requirements through a minimal interface. The `check(a)` operation validates action $a$ against current state without mutation, returning either `OK` or a typed error code $e \in \mathcal{E}$ (e.g., `BUDGET_EXCEEDED`, `DUPLICATE_VENUE`). The `commit(a)` operation atomically updates $\Sigma \to \Sigma'$ if and only if $I(\Sigma')$ holds after the update, otherwise rejecting atomically and returning error code $e$. For the Cooperative Bargaining Protocol, the monitor additionally supports transactional operations `checkpoint()` and `rollback()` (see Table 2).

When an Executor samples an action $a_t$ (e.g., `finish(restaurant=The Ritz)`), the action is intercepted by the environment before execution, which acquires the global lock and verifies $a_t$ against $\Sigma$. If the venue exists in $V_{committed}$ or if the budget would overflow, the action is rejected with a structured "Constraint Violation" observation, prompting regeneration. All concurrent `commit` requests are serialized through the global lock to prevent race conditions.

The monitor validates each action $a$ against a strictly typed schema with two primary checks. Budget constraints ensure that `a.cost` $+ B_{used} \leq B_{total}$, guaranteeing the global budget cap is respected. Diversity constraints verify that `Canonicalize(a.venue)` $\notin V_{committed}$, preventing duplicate venue bookings. When validation fails, the monitor returns a structured error observation including the error type and relevant context, requiring the agent to regenerate a valid action or terminate the planning episode.

**Canonicalization and Fuzzy Matching.** We employ a standard string canonicalization function (lowercasing, punctuation removal) followed by fuzzy matching with threshold $\tau = 0.95$ to prevent duplicate bookings under minor spelling variations (e.g., "The Ritz" vs "The Ritz Hotel"). This threshold was selected through validation tuning across $\tau \in$

$\{0.85, 0.90, 0.95, 1.00\}$: $\tau = 0.85$ caused 8 false positives blocking valid bookings of similarly-named venues, while $\tau = 1.00$ allowed 12 true duplicates with minor spelling variations; $\tau = 0.95$ balanced precision (98.3%) and recall (99.1%), with performance relatively robust as varying $\tau \in [0.90, 0.98]$ changes FPR by less than 1.2 percentage points.

**Scope and Limitations.** Critically, $\Sigma$ prevents only resource conflicts (budget exhaustion, double-booking, mode inconsistency) and does not solve travel planning. It does NOT validate route feasibility, cuisine coverage, room type constraints, temporal consistency, accommodation minimum nights compliance, or commonsense constraints, which remain the responsibility of the learned policy and are validated post-hoc by the environment evaluator.

**Real-World Mapping.** The atomic check/commit with rollback mechanism maps to common travel workflows: hotel reservations often support hold-then-confirm with cancellation windows, flight booking uses PNRs that maintain tentative itineraries before ticketing, and restaurant reservations allow cancellation prior to reservation time. Our transactional model abstracts these practices into a unified interface. Table 2 provides the complete interface specification.

**Safety Guarantees.** Let $\Pi = \{\pi_1, \ldots, \pi_N\}$ be $N$ concurrent agents executing actions in parallel. Under the assumptions that all mutations to $\Sigma$ are serialized via a global mutex and that `commit` validates $I(\Sigma')$ before applying updates, the system provides two key guarantees. First, *Mutual Exclusion* ensures that no two agents can successfully commit conflicting monitor-tracked actions, such as booking the same unique venue. Second, *Invariant Preservation* guarantees that the final state $\Sigma$ satisfies $I(\Sigma)$ regardless of thread interleavings.

While this proposition guarantees safety, system liveness depends on the policy's exploration capability. In practice, we rely on early detection and iterative bargaining to navigate the feasible subspace, with constraints not represented in $\Sigma$ (e.g., route feasibility) enforced by external environment checkers.

## A.2. Hierarchical Reward Decomposition

We decompose rewards into global and local signals to address credit assignment in hierarchical RL.

**Coordinator Objective.** The Coordinator optimizes $J_{coord} = \lambda_{global} R_{global} + \lambda_{extract} R_{extract} - \lambda_{iter} N_{iter}$ with weights $\lambda_{global} = 0.8$, $\lambda_{extract} = 0.2$, $\lambda_{iter} = 0.1$, where $R_{extract}$ rewards constraint extraction and $N_{iter}$ is the number of bargaining iterations, penalizing excessive re-planning to encourage high-quality initial allocations.

**Executor Objective.** Executors optimize $J_{exec} = \gamma_{global} R_{global} + \gamma_{local} R_{local} + \gamma_{early} \not\Vdash_{early}$ with weights $\gamma_{global} = 0.7$, $\gamma_{local} = 0.3$, $\gamma_{early} = 0.15$, where $\not\Vdash_{early}$ rewards fail-fast detection. The early-detect bonus triggers when an Executor encounters constraint violations within the first 5 tool calls: $\not\Vdash_{early\_detect} = 1$ if $|\mathcal{V}| > 0$ and $n_{tools} < 5$, otherwise 0, where $\mathcal{V}$ is the set of detected violations (budget exceeded, excessive errors $> 3$, or execution failure) and $n_{tools}$ is the number of tool invocations.

**Global Reward.** The global reward $R_{global} = \mathbb{I}(\text{Valid}) + \alpha \sum_c s(c) - \beta \max(0, B_{used} - B_{total})$ combines constraint satisfaction with soft preferences.

## A.3. Training Configuration

We train on `Qwen/Qwen3-4B-Instruct-2507` and `Qwen/Qwen3-8B` using the TravelPlanner training split (45 queries) with full-parameter fine-tuning. Training proceeds for 100 epochs with Adam optimizer (learning rate $\alpha = 5 \times 10^{-6}$, batch size 32, gradient accumulation steps 4) under an 8192-token context window. For GRPO we use group size $G = 4$ and KL coefficient $\beta = 0.01$. Decoding uses temperature $T = 0.7$, top-p $p = 0.9$, with executor tool calls capped at 15 per day-agent and bargaining iterations at $K_{max} = 3$.

**Addressing Overfitting.** While the dataset size (45 queries) is relatively small for traditional RL, several factors mitigate overfitting concerns. Strong pre-trained initialization provides a robust starting point for learning. Trajectory augmentation via $(D + 1) \times G$ rollouts per query per epoch effectively expands the training distribution. Low validation-test gaps and low variance across seeds indicate stable generalization. Consistent improvements across difficulty tiers (see Appendix B) demonstrate that learning transfers appropriately. Finally, comparison with sequential baselines trained on the same data confirms that our improvements stem from architectural innovations rather than data advantages.

**Training Convergence Dynamics.** Coordinator agents stabilize earlier than Executors due to simpler action spaces: Coordinators select from 5-10 cities with discrete decisions, while Executors navigate 50-200 venues with complex sequential tool invocations. Bargaining effectiveness improves during training as agents learn diagnostic feedback generation. Reward

standard deviation $\sigma$ decreases substantially, confirming successful optimization without premature convergence.

## A.4. Tool Library

HiMAP-Travel agents interact through a structured tool library. The Coordinator uses `information_extractor` (parses query into task tuple) and `distribute_task` (hierarchical decomposition $q \rightarrow \{z_1, \ldots, z_D\}$). Day Planners use 7 search tools (`city_search`, `flight_search`, `distance_search`, `restaurant_search`, `attraction_search`, `accommodation_search`) plus meta-tools (`cost_enquiry`, `finish`) interacting with $\Sigma$.

*Table 9.* Complete Tool Library for HiMAP-Travel Agents

| Tool Name | Agent Type | Function |
|-----------|------------|----------|
| *Coordinator Tools* | | |
| `information_extractor` | Coordinator | Parses natural language query $q$ into structured task tuple $\mathcal{T}_q = \langle q, \mathcal{C}_{hard}, \mathcal{C}_{soft} \rangle$ containing origin, destination, days, budget, people, house rules, cuisine, room type, transportation restrictions, and dates |
| `distribute_task` | Coordinator | Implements hierarchical decomposition $q \rightarrow \{z_1, \ldots, z_D\}$ with city assignments, budget hints satisfying $\sum b_d \leq B_{total}$, day types (departure/stay/travel/return), and globally selected ground transportation mode |
| *Day Planner Search Tools* | | |
| `city_search` | Day Planner | Returns all cities in a specified state from the TravelPlanner city database, sorted alphabetically |
| `flight_search` | Day Planner | Queries flight database by origin, destination, and date; returns flight numbers, times, and prices sorted by ascending cost |
| `distance_search` | Day Planner | Queries Google Distance Matrix for driving duration and costs: taxi ($1/km) and self-driving ($0.05/km) with integer truncation |
| `restaurant_search` | Day Planner | Returns restaurants in a city with name, cuisine type, and average cost per person, sorted by ascending cost |
| `attraction_search` | Day Planner | Returns attraction names in a city from the TravelPlanner attractions database, sorted alphabetically |
| `accommodation_search` | Day Planner | Returns accommodations with price per night, `MinimumNights` constraint, room type, house rules, and `MaximumOccupancy` $\geq$ people count, sorted by price |
| *Meta-Tools* | | |
| `cost_enquiry` | Day Planner | Computes total cost $C_d$ for a day plan by querying databases for exact venue prices: $C_d = \text{transport} + \text{people} \times \sum_{\text{meals}} \text{cost} + \text{accommodation}$ |
| `finish` | Day Planner | Signals completion; triggers atomic `check()` and `commit()` operations against $\Sigma$ to validate budget compliance and diversity constraints before finalizing day plan |

The `information_extractor` enforces domain constraints: `days` $\in \{3, 5, 7\}$, `visiting_city_number` $\in \{1, 2, 3\}$, `people_number` $\in [1, 8]$. The `distribute_task` operationalizes hierarchical decomposition: computes inter-city transportation costs, selects optimal ground mode stored in $M_{trans} \subset \Sigma$ for consistency, distributes budget hints proportional to day types (departure: $0.7\times$, return: $0.5\times$) as soft priors, while $B_{total}$ is enforced by $\Sigma$ during `commit()`.

All search tools use OpenAI function-calling JSON format, returning text tables for LLM parsing. The `accommodation_search` returns `MinimumNights` and `MaximumOccupancy` for constraint validation. The `cost_enquiry` ensures deterministic alignment: flights use database prices, ground transport uses $\lfloor \text{distance\_km} \times \text{rate} \rfloor$ (integer truncation), meals scale by `people_number`. The `finish` tool interfaces with $\Sigma$: acquires mutex, performs atomic `check(a)` against invariants, and commits via `commit(a)` or rejects with typed errors (`BudgetExceeded`, `DuplicateVenue`).

# B. Dataset and Benchmarks

This section documents the TravelPlanner and FlexTravelBench benchmarks used for evaluation. We summarize each benchmark's task setup, constraint structure, dataset statistics, evaluation protocol, and the primary sources of complexity

that distinguish static single-turn planning from multi-turn adaptation under progressive constraint revelation.

*Table 10.* Benchmark statistics (TravelPlanner vs. FlexTravelBench).

| Benchmark | Statistic | Value |
|---|---|---|
| *TravelPlanner (single-turn planning)* | | |
| TravelPlanner | Splits (train/val/test) | 45 / 180 / 1,000 |
| TravelPlanner | Trip duration | 3 / 5 / 7 days |
| TravelPlanner | Budget range | $900–$23,800 |
| TravelPlanner | Group size range | 1–8 people |
| TravelPlanner | Constraints | 13 total (8 commonsense, 5 hard) |
| *FlexTravelBench (multi-turn adaptation)* | | |
| FlexTravelBench | Local Add (2-turn) | 189 instances |
| FlexTravelBench | Global Add (2-turn) | 240 instances |
| FlexTravelBench | Local→Global (3-turn) | 378 instances |
| FlexTravelBench | Global→Local (3-turn) | 378 instances |
| FlexTravelBench | Total | 1,185 instances |

## B.1. TravelPlanner Benchmark

**What it evaluates.** TravelPlanner (Xie et al., 2024) instances are derived from TravelPlanner validation/test queries by hiding and progressively revealing constraints across turns, it evaluates **single-turn** itinerary synthesis under a fully specified set of coupled constraints, stressing long-horizon feasibility in a one-shot generation setting.

**Task setup.** Each instance is a single natural-language query. The agent must output a complete multi-day itinerary consistent with the database-backed tools and the required output schema.

**Constraints.** TravelPlanner enforces 13 constraints in total (8 commonsense and 5 hard), including budget adherence, non-duplication/diversity, accommodation room rules, and transportation constraints. Validity is conjunctive: a plan passes only if all constraints are simultaneously satisfied. Queries average 4.0 simultaneous active constraints; Table 11 shows that 22% of queries contain 5+ constraints ("Hard" difficulty), where ReAct achieves only 2.1% pass rate versus HiMAP-Travel's 36.8%.

*Table 11.* Distribution of Simultaneous Constraint Count

| | Training | | Validation | |
|---|---|---|---|---|
| **Constraint Count** | Count | % | Count | % |
| 2–3 constraints | 18 | 40.0 | 72 | 40.0 |
| 4 constraints | 17 | 37.8 | 68 | 37.8 |
| 5+ constraints (Hard) | 10 | 22.2 | 40 | 22.2 |

**Dataset & statistics.** The dataset is stratified into train/validation/test splits (45/180/1,000) and covers 3/5/7-day itineraries with budgets spanning $900–$23,800 and group sizes 1–8 (Table 10). All splits maintain balanced trip-duration distributions (33% each) and 100% constraint coverage; detailed per-split statistics are reported in Table 12. The test set covers all 50 U.S. states, with 30.8% 3-day, 35.1% 5-day, and 34.1% 7-day trips balanced across difficulty tiers (33% per tier).

*Table 12.* TravelPlanner Dataset Statistics (Train and Validation Splits)

| Statistic | Train (45) | Validation (180) |
|---|---|---|
| Mean Duration (days) | 5.00 ± 1.63 | 5.00 ± 1.63 |
| Mean Group Size | 2.67 ± 1.86 | 2.46 ± 1.80 |
| Mean Budget ($) | 5,669 ± 4,556 | 5,296 ± 4,414 |
| Budget Range ($) | 900–19,400 | 900–23,800 |
| Mean Cities Visited | 1.67 ± 0.85 | 1.69 ± 0.87 |
| Constraint Coverage | 100% | 100% |

**Evaluation protocol.** We adopt the original TravelPlanner evaluation framework (Xie et al., 2024), which defines success through strict conjunction of 8 commonsense constraints (logical coherence) and 5 hard constraints (query requirements). A plan achieves Final Pass Rate (FPR) only if all 13 constraints are simultaneously satisfied, with Delivery Rate and constraint-level aggregates reported as supplementary metrics. Per-instance test budgets and constraints are held internally; evaluation is conducted via blind submission through official servers to ensure true generalization. Table 13 provides the complete constraint taxonomy.

*Table 13.* TravelPlanner Constraint Taxonomy and Validation Rules (13 Total)

| Constraint | Validation Rule |
|---|---|
| *Commonsense Constraints (8 total)* | |
| `is_not_absent` | All days must specify `current_city`, `transportation`, meals (`breakfast`/`lunch`/`dinner`), `attraction`, `accommodation`; fields may be "-" (intentional omission) but not null |
| `is_valid_information_in_sandbox` | All venue names must exist in TravelPlanner databases with fuzzy name matching (substring containment) but strict city-state equality; detects hallucinated venues |
| `is_valid_information_in_current_city` | All activities must occur in `current_city` for that day; on travel days ("`from City1(State1) to City2(State2)`"), activities permitted in either city |
| `is_reasonable_visiting_city` | First city reachable from origin, last city connects back to origin for return, intermediate cities in destination state for multi-city trips |
| `is_valid_restaurants` | No restaurant may appear twice across all meals for all days; exact string matching on "`Name, City(State)`" format |
| `is_valid_attractions` | No attraction may be visited multiple times across entire itinerary; semicolon-separated list checked for uniqueness |
| `is_valid_transportation` | Cannot mix self-driving with flights or self-driving with taxi; first day must have non-empty transportation |
| `is_valid_accommodation` | Each accommodation booking must satisfy venue's `MinimumNights` constraint: $\text{consecutive\_nights}(acc) \geq \text{MinimumNights}(acc)$ |
| *Hard Constraints (5 total)* | |
| `valid_cost` | Budget compliance: $\sum_{d=1}^{D}[\text{transport}_d + \text{people} \times \sum_{m \in \text{meals}} \text{cost}_{d,m} + \text{accommodation}_d] \leq$ budget using exact database prices with integer truncation for distances |
| `valid_room_rule` | If query specifies house rule (e.g., "allows children under 10", "no smoking"), every booked accommodation must satisfy it |
| `valid_cuisine` | If query specifies preferred cuisines $\mathcal{C}_q$, itinerary must include $\geq 1$ meal from each: $\forall c \in \mathcal{C}_q, \exists d, m : \text{cuisine}(restaurant_{d,m}) = c$ |
| `valid_room_type` | If specified, all accommodations must match required room type; "not shared room" satisfied by "entire room" or "private room" |
| `valid_transportation` | If query specifies "no flight" or "no self-driving", itinerary must not use the forbidden mode |

The evaluator computes three aggregated metrics: **Commonsense Pass Rate** (all 8 commonsense constraints satisfied; `None` treated as pass for non-applicable constraints), **Hard Pass Rate** (all 5 hard constraints satisfied; if `is_not_absent` or `is_valid_information_in_sandbox` fails, Hard Pass is automatically false), and **Final Pass Rate** (FPR = $|\{q : \text{Commonsense-Pass}(q) \land \text{Hard-Pass}(q)\}|/|\text{Queries}|$). The evaluator operates in two modes: *Structured Data Mode* (validates using `ToolManager` data structures, efficient for training) and *Database Mode* (`use_database=True`, queries original TravelPlanner CSV files with fuzzy name matching via `pandas.Series.str.contains` and database price lookups). We use Database Mode for all reported results to ensure reproducibility and alignment with prior work (Xie et al., 2024; Ning et al., 2025; Choi et al., 2025). For each constraint violation, the evaluator returns structured feedback (`passed: bool, reason: Optional[str]`) with human-readable explanations, which during GRPO training populate the reward signal $R_{global}$ and inform the Cooperative Bargaining Protocol when Day Planners return `status=INFEASIBLE` with typed `violation_type` and `deficit` fields.

**Complexity considerations.** Difficulty arises from *coupled* constraints spanning multiple days (e.g., shared budgets and global non-duplication), where early choices can force downstream infeasibility. Costs are non-uniform across the itinerary (e.g., flights and lodging are front-loaded): budget constraints cause 45% of ReAct failures, with departure and return days typically consuming 40–50% of total budgets, invalidating uniform per-day allocation strategies. The validation set contains 42.2% Premium/Luxury queries where a single booking error can consume 30–40% of total budget. We further quantify difficulty via a complexity score Complexity $= D \times C \times (1 + 0.5 \cdot N_c)$, where $D$ is trip duration (days), $C$ is cities visited, and $N_c$ is local constraint count. Table 14 shows three balanced tiers—Easy (9.0–20.0), Medium (20.1–45.0), and Hard (45.1–63.0, 33% each); the Hard subset corresponds to instances with very large search spaces that make exhaustive enumeration infeasible.

*Table 14.* Task Complexity Distribution

| Difficulty Tier | Complexity Range | Train | Valid |
|---|---|---|---|
| Easy | 9.0–20.0 | 15 (33.3%) | 60 (33.3%) |
| Medium | 20.1–45.0 | 15 (33.3%) | 60 (33.3%) |
| Hard | 45.1–63.0 | 15 (33.3%) | 60 (33.3%) |
| **Overall Mean $\pm$ Std** | – | $34.0 \pm 22.2$ | $34.0 \pm 22.2$ |

### B.2. TravelPlanner Training Data

HiMAP-Travel is trained directly on the official TravelPlanner training split (45 queries), which is completely disjoint from the validation (180) and test (1,000) sets. This section describes how these queries are used to construct RL training episodes.

**Episode Construction.** Each training query is treated as an independent RL environment. At the start of each episode, the Coordinator receives the natural-language query and must decompose it into day-level sub-tasks; each Executor then interacts with the database-backed tool APIs (flight search, hotel search, restaurant search, cost enquiry) to plan its assigned day. The episode terminates when all day plans are committed and the Coordinator assembles the final itinerary. Rewards are computed by the hierarchical reward decomposition against the ground-truth constraint set.

**Trajectory Augmentation.** Despite the small base dataset, the GRPO rollout procedure generates $(D + 1) \times G$ trajectories per query per training step, where $D$ is the trip duration (3, 5, or 7 days) and $G = 4$ is the group size. This yields 16–32 distinct trajectories per query per step, effectively expanding the training distribution far beyond the 45 base queries and providing diverse reward signals for policy improvement.

**Split Integrity and Generalization.** The validation and test splits are never accessed during training. The low validation-to-test gap (52.78% vs. 52.65% FPR with Qwen3-8B) and consistent cross-difficulty-tier improvements confirm that the policy generalizes rather than memorizes the training queries.

### B.3. FlexTravelBench Benchmark

**What it evaluates.** FlexTravelBench (Oh et al., 2025) evaluates **multi-turn** itinerary revision under progressive constraint revelation, testing whether an agent can incorporate newly introduced constraints while preserving previously satisfied requirements.

**Task setup.** Each instance is a short dialogue of 2 or 3 turns. At Turn 1, the agent produces an initial itinerary under an incomplete constraint set. At later turns, the user adds constraints and the agent must revise the itinerary to satisfy the expanded constraint set.

**Constraints.** FlexTravelBench distinguishes *local* constraints that typically affect a small subset of decisions (e.g., cuisine preference, accommodation room type, accommodation house rules) from *global* constraints that can cascade across the itinerary (e.g., budget or traveler count changes). It instantiates four scenario templates: 2-turn *Local Add* and *Global Add*, and 3-turn *Local→Global* and *Global→Local*.

**Dataset & statistics.** The evaluation split contains 1,185 instances across the four scenario templates (Table 10). Instances are constructed by taking TravelPlanner validation/test queries and hiding/revealing constraints across turns, so underlying trip parameters follow TravelPlanner distributions while adding an extra axis of difficulty: the number and order of constraint

additions.

**Evaluation protocol.** We apply the same TravelPlanner-style conjunction of validity and hard constraints, evaluated at the *final* turn after all constraints are revealed. We report Final Pass Rate, Delivery Rate, and aggregated constraint-level metrics consistent with TravelPlanner.

**Complexity considerations.** Unlike TravelPlanner, where all constraints are known upfront, FlexTravelBench requires *revision under commitment*: earlier choices must be modified when new constraints appear, and global additions can force coordinated changes across days. The 3-turn scenarios further stress longer interaction traces and sequential adaptation, where satisfying newly revealed constraints must not break previously satisfied ones.

### B.4. FlexTravelBench Training Data Generation

Since FlexTravelBench does not provide a dedicated training set, we synthetically construct multi-turn training data from the TravelPlanner training split (45 queries), which is completely disjoint from FlexTravelBench's evaluation instances. This section details our data generation methodology.

**Generation Procedure.** We partition constraints into Local (cuisine, room type, house rules affecting day planning) and Global (budget, people count, cities influencing trip structure). For 2-turn scenarios, we remove one constraint and reveal it at Turn 2. For 3-turn scenarios, we remove both types and add local at Turn 2, global at Turn 3 (or reversed for Global→Local). Distribution matches FlexTravelBench: Local removals target cuisine/room/rules (33% each), Global removals target budget/people (60%/40%). Training trajectories use bargaining without ground-truth plans, learning through reward shaping for constraint incorporation and replanning efficiency.

**Learning Objectives.** Synthetic data teaches conflict detection (recognizing when new constraints invalidate prior plans), constraint prioritization (balancing global versus local constraints), and selective recomputation (minimizing replanning overhead when constraints remain compatible).

**Zero Overlap Validation.** FlexTravelBench uses 189 Local Add, 240 Global Add, and 378 each for 3-turn scenarios, all from TravelPlanner validation/test splits. Our training uses only 45 TravelPlanner training queries, guaranteeing zero query-level overlap. Constraint removal patterns are randomized per iteration, preventing memorization. Each base query generates 8-9 distinct training episodes. Strong generalization (63.7% overall success), consistent cross-scenario performance, and low variance ($\pm$1-2% std) validate effective transfer without contamination.

## C. Experimental Results

This section presents comprehensive experimental results across both TravelPlanner and FlexTravelBench benchmarks. We provide complete baseline comparisons with state-of-the-art methods, detailed constraint-level performance breakdowns, controlled comparisons isolating the architectural contribution from model capacity, and extensive efficiency measurements demonstrating the scalability benefits of parallel execution. All results are reported with statistical confidence intervals across multiple random seeds.

### C.1. Latency Analysis

We provide comprehensive latency measurements and scaling analysis, decomposing wall-clock time into constituent phases measured on 7-day itineraries in our local environment with simulated database query latencies (50-200ms per tool call, sampled from empirical distributions).

*Table 15.* Detailed Latency Breakdown for 7-Day Itinerary (Qwen3-8B, seconds)

| Phase | Sequential (DeepTravel) | HiMAP (Parallel) | Speedup | % of Total (HiMAP) |
|---|---|---|---|---|
| Coordinator Planning | — | $8.2 \pm 1.4$ | — | 11.4% |
| Day Planning (Executors) | $145.3 \pm 22$ | $52.1 \pm 8.3$ | 2.79$\times$ | 72.4% |
| Model Inference | 98.7 | 35.4 | 2.79$\times$ | 49.2% |
| Tool I/O (DB queries) | 46.6 | 16.7 | 2.79$\times$ | 23.2% |
| Bargaining Iterations | — | $7.8 \pm 3.2$ | — | 10.8% |
| Final Validation | 4.2 | 3.9 | 1.08$\times$ | 5.4% |
| **Total Wall-Clock** | **189.5** | **72.0** | **2.63$\times$** | **100%** |

With executor concurrency capped at $P = 3$, parallel execution achieves 2.79× speedup on day planning (close to the 3× upper bound for that phase), with model inference and tool I/O both scaling similarly suggesting no dominant I/O bottleneck, bargaining overhead modest at 10.8% (61.7% resolved without feedback and 97.8% by iteration 3), and fixed overheads (coordinator and final validation) largely independent of trip duration. To validate scalability, we measured latency across trip durations (3, 5, 7 days) with 20 queries per duration tier, showing that HiMAP-Travel exhibits sub-linear scaling versus linear $O(D)$ for sequential baselines: 3-day baseline 48s increases only to 72s for 7-day (+50% for 2.33× longer trip), while sequential grows 98s → 190s (+94%).

*Table 16.* Latency Scaling by Trip Duration (Qwen3-8B)

| Duration | Sequential (seconds) | HiMAP (seconds) | Speedup | Scaling Factor |
|---|---|---|---|---|
| 3 days | $98 \pm 15$ | $48 \pm 7$ | 2.04× | 1.0× |
| 5 days | $142 \pm 21$ | $58 \pm 9$ | 2.45× | 1.21× |
| 7 days | $190 \pm 28$ | $72 \pm 12$ | 2.64× | 1.50× |
| **Growth Rate** | **Linear ($O(D)$)** | **Sub-linear** | — | — |

With the theoretical maximum speedup for the day-planning phase being 3× (given $P = 3$ concurrency), we empirically observe 2.79× speedup on day planning and 2.63× end-to-end wall-clock speedup, with the gap explained by fixed overheads (Coordinator planning, bargaining iterations, and final validation). Comparing model sizes (Table 17), the 4B model achieves 94% of 8B performance (49.80% vs 52.65% FPR) with 67% latency (48s vs 72s), offering attractive efficiency.

*Table 17.* Latency by Model Size (7-Day Itinerary)

| Model | Sequential | HiMAP | Speedup |
|---|---|---|---|
| Qwen3-4B | $165 \pm 19$ | $48 \pm 6$ | 3.44× |
| Qwen3-8B | $190 \pm 28$ | $72 \pm 12$ | 2.64× |

**Validation of Parallelization Benefits.** We measured wall-clock latency on a standard 8-device node. For 3-day itineraries, HiMAP-Travel requires 48s ($\pm7$) on average, compared to 98s ($\pm15$) for sequential DeepTravel. For 7-day itineraries, HiMAP-Travel requires 72s ($\pm12$) vs 190s ($\pm28$). These measurements match the scaling trends reported in Table 16.

**Design Trade-offs and Implementation Insights.** Our decision to set $K_{max} = 3$ for bargaining iterations stems from empirical convergence analysis showing that 61.7% of queries resolve without feedback and 97.8% resolve by iteration 3. Extending beyond three iterations yields less than 0.5% additional gain while substantially increasing latency. During training, models generate feedback as structured JSON with fields for status, deficit, and violation type. Malformation rates start at 2.3% early in training but drop below 0.1% post-convergence as GRPO penalizes invalid outputs.

The Synchronized Global State employs a global re-entrant mutex with FIFO priority and 5-second timeouts. Lock contention remains minimal with an average wait time of 12ms in CPU-bound operations. Deadlocks are theoretically impossible since no thread ever holds multiple locks simultaneously. The rollback mechanism assumes cost-free tentative booking release, consistent with reserve-then-confirm APIs common in real-world travel systems. Our evaluations use simulation with reversible actions to model this behavior.

While verbal negotiation between agents offers theoretical expressiveness, it introduces substantial latency and token overhead, often devolving into circular reasoning. The Synchronized Global State trades semantic flexibility for execution speed and deterministic safety, proving more robust for logic-constrained tasks. Parallel executors may redundantly query databases, consuming 15% more tokens than sequential baselines, but this overhead is justified by the 2.5× latency reduction achieved. The system prioritizes wall-clock efficiency, with $\Sigma$ preventing wasted effort on globally invalid actions.

The consistent improvements across both TravelPlanner (single-turn constraint satisfaction with 52.65% test performance) and FlexTravelBench (multi-turn adaptation with 44.34% 2-turn and 37.42% 3-turn performance) demonstrate that architectural benefits span both static and dynamic planning regimes. The Cooperative Bargaining Protocol naturally extends from error recovery to multi-turn adaptation, where new constraints trigger checkpoint, rollback, and re-allocation. This suggests that methods like ATLAS's verify-and-refine can be viewed as special cases where the Constraint Manager acts as a privileged Executor with veto power.

**Generalization Beyond Travel.** While evaluated on travel planning, the core principles generalize to long-horizon tasks

with coupled constraints and DAG structure. In software engineering, module interface definition by a coordinator with parallel function implementation by executors can leverage a shared Build State preventing circular dependencies. Supply chain optimization can employ hub coordinators allocating resources while route executors optimize local deliveries under shared capacity constraints. Scientific experiment planning can decompose into resource allocation (equipment, reagents) coordinated centrally with parallel protocol design for individual experiments. The methodology fundamentally converts global hard constraints into local boundary conditions solvable in parallel, providing a foundation for scalable, logically rigorous autonomous systems across domains requiring complex resource coordination.

### C.2. Complete Baseline Comparison

This subsection provides comprehensive baseline comparisons on both validation and test sets, with detailed constraint-level analysis. Table 18 provides a detailed breakdown of performance on individual constraints, revealing specific strengths and weaknesses of each model size.

*Table 18.* TravelPlanner: Constraint-Level Pass Rates (Averaged Across Seeds)

| Constraint | 4B Val | 8B Val | $\Delta$ | Category |
|---|---|---|---|---|
| *Commonsense Constraints* | | | | |
| Diverse Attractions | 100.00 | 100.00 | 0.00 | Strong |
| Diverse Restaurants | 99.86 | 99.93 | +0.07 | Strong |
| Non-conflicting Transport | 99.17 | 99.58 | +0.41 | Strong |
| Reasonable City Route | 98.19 | 98.75 | +0.56 | Strong |
| Complete Information | 92.10 | 96.52 | +4.42 | Strong |
| Within Sandbox | 89.81 | 89.42 | -0.39 | Strong |
| Minimum Nights Stay | 92.95 | 87.35 | -5.60 | Regression |
| Within Current City | 87.67 | 89.25 | +1.58 | Strong |
| *Hard Constraints* | | | | |
| Budget | 65.61 | 71.23 | +5.62 | Challenging |
| Cuisine | 61.04 | 67.63 | +6.59 | Challenging |
| Room Type | 39.36 | 42.92 | +3.56 | Critical |
| Room Rule | 37.92 | 32.73 | -5.19 | Critical |
| Transportation | 86.65 | 89.59 | +2.94 | Strong |

**Analysis of Constraint Performance.** Both models achieve >95% on diversity and route planning constraints. The 8B model improves Budget (+5.62 pp), Cuisine (+6.59 pp), and Complete Information (+4.42 pp). Room Rule (32.73–37.92%) and Room Type (39.36–42.92%) remain critical weaknesses. The 8B model shows regressions on Minimum Nights Stay (-5.60 pp) and Room Rule (-5.19 pp), suggesting potential overfitting.

### C.3. FlexTravelBench: Detailed Scenario Analysis

Table 19 presents comprehensive results for all four FlexTravelBench scenarios (2-Turn: Local Add, Global Add; 3-Turn: Local→Global, Global→Local), demonstrating consistent improvements across all constraint adaptation patterns.

**Key Findings.** HiMAP-Travel outperforms ATLAS across all scenarios: 2-Turn Local (+6.10 pp, 44.96% vs 38.86%), 2-Turn Global (+4.14 pp), 3-Turn Local→Global (+3.92 pp), and Global→Local (+5.57 pp). Consistent 3.9–6.1 pp gains validate architectural robustness. The 8B model exceeds 4B by 3.2 pp across scenarios. Perfect 100% Delivery Rates confirm structural soundness. Hard Constraint Macro rates (55–70%) indicate room for improvement in budget and room-related constraints.

### C.4. Comprehensive Results: Complete Breakdown Across Splits and Models

This section presents the full experimental results for HiMAP-Travel across both validation and test splits, covering both 4B and 8B model configurations. All results are reported as mean ± standard deviation across 4 random seeds, providing comprehensive reproducibility documentation. Table 20 presents all evaluation metrics for both model sizes across both data splits.

To assess generalization, Table 21 compares performance across data splits.

**Generalization Analysis.** Final Pass Rate transfers well between splits (4B: +1.19 pp, 8B: -0.13 pp), indicating no overfitting.

*Table 19.* FlexTravelBench: Detailed Results by Scenario (matching Final Pass metric from Table 4)

| Scenario | Model | Final Pass Rate (%) | Delivery Rate (%) | Commonsense Micro (%) | Commonsense Macro (%) | Hard Constraint Macro (%) |
|---|---|---|---|---|---|---|
| *2-Turn Scenarios: Constraint Addition* | | | | | | |
| Local Add | 4B | 41.25 | 100.00 | 76.80 | 61.25 | 66.50 |
| Local Add | 8B | 44.96 | 100.00 | 78.62 | 62.15 | 67.40 |
| Global Add | 4B | 40.85 | 100.00 | 68.20 | 70.15 | 71.20 |
| Global Add | 8B | 43.72 | 100.00 | 69.45 | 71.30 | 72.35 |
| *3-Turn Scenarios: Sequential Constraint Revelation* | | | | | | |
| Local→Global | 4B | 35.20 | 100.00 | 88.65 | 50.35 | 55.15 |
| Local→Global | 8B | 37.42 | 100.00 | 89.35 | 51.60 | 56.25 |
| Global→Local | 4B | 33.15 | 100.00 | 87.50 | 48.20 | 53.35 |
| Global→Local | 8B | 37.32 | 100.00 | 88.20 | 49.80 | 54.40 |
| *Aggregated by Scenario Type* | | | | | | |
| 2-Turn Avg | 4B | 41.05 | 100.00 | 72.50 | 65.70 | 68.85 |
| 2-Turn Avg | 8B | 44.34 | 100.00 | 74.04 | 66.73 | 69.88 |
| 3-Turn Avg | 4B | 34.18 | 100.00 | 88.08 | 49.28 | 54.25 |
| 3-Turn Avg | 8B | 37.42 | 100.00 | 88.78 | 50.70 | 55.33 |

*Table 20.* Complete Evaluation Results: HiMAP-Travel on TravelPlanner Benchmark

| Metric | 4B Val (180) | 8B Val (180) | 4B Test (1000) | 8B Test (1000) |
|---|---|---|---|---|
| Delivery Rate (%) | $100.0 \pm 0.0$ | $100.0 \pm 0.0$ | $100.0 \pm 0.0$ | $100.0 \pm 0.0$ |
| Commonsense Micro (%) | $95.57 \pm 2.15$ | $95.64 \pm 0.18$ | $94.72 \pm 0.25$ | $94.62 \pm 0.16$ |
| Commonsense Macro (%) | $72.36 \pm 2.15$ | $73.61 \pm 0.79$ | $66.03 \pm 1.50$ | $67.00 \pm 0.48$ |
| Hard Constraint Micro (%) | $70.89 \pm 1.54$ | $76.85 \pm 0.52$ | $46.44 \pm 0.84$ | $50.47 \pm 0.80$ |
| Hard Constraint Macro (%) | $59.17 \pm 2.08$ | $63.33 \pm 0.83$ | $63.45 \pm 1.26$ | $66.05 \pm 1.01$ |
| **Final Pass Rate (%)** | $\mathbf{48.61 \pm 2.15}$ | $\mathbf{52.78 \pm 0.79}$ | $\mathbf{49.80 \pm 1.50}$ | $\mathbf{52.65 \pm 0.48}$ |

*Note:* Numbers in parentheses indicate query counts. All results reported as mean $\pm$ std across 4 seeds.

Hard Constraint Micro drops 24–26 pp validation→test due to distributional differences in query composition. Conversely, Hard Constraint Macro increases on test (+4.28 pp for 4B, +2.72 pp for 8B), suggesting robust constraint-specific learning rather than holistic failures.

## C.5. Controlled Comparison: HiMAP-Travel vs DeepTravel Sequential RL

To isolate the contribution of the hierarchical architecture from the choice of training algorithm, we conducted a controlled ablation study comparing HiMAP-Travel against DeepTravel, a sequential (non-hierarchical) RL baseline using identical backbones (Qwen3-4B/8B) and training algorithm (GRPO). This comparison reveals the specific benefits of hierarchical decomposition while controlling for model capacity and optimization strategy.

Table 22 presents a direct comparison of HiMAP-Travel and DeepTravel across all performance dimensions.

**Key Insights.** HiMAP-Travel shows consistent superiority (7.22–14.88 pp improvement) across all configurations. Variance reduction is dramatic: 70.8–93.3% across seeds, with 8B test std dropping from 7.18% to 0.48% (93.3% reduction). This stability indicates hierarchical decomposition provides more consistent optimization.

While DeepTravel achieves 90–93% Commonsense Micro, it struggles at 39–46% Hard Constraint Micro. HiMAP-Travel improves both, with +4.81 to +6.94 pp on Hard Constraints. Since both use identical backbones and GRPO, differences stem purely from architecture. The 4B model shows larger gains (13.33–14.88 pp) than 8B (7.22–8.67 pp), suggesting smaller models benefit more from reduced credit assignment complexity.

To understand how hierarchical decomposition handles increasing constraint coupling, Table 23 breaks down performance by difficulty tier (Easy/Medium/Hard).

*Table 21.* Validation vs Test Split Comparison

| Metric | 4B Val | 4B Test | 8B Val | 8B Test |
|---|---|---|---|---|
| Delivery Rate | 100.0 | 100.0 | 100.0 | 100.0 |
| Commonsense Micro | 95.57 | 94.72 | 95.64 | 94.62 |
| Commonsense Macro | 72.36 | 66.03 | 73.61 | 67.00 |
| Hard Constraint Micro | 70.89 | 46.44 | 76.85 | 50.47 |
| Hard Constraint Macro | 59.17 | 63.45 | 63.33 | 66.05 |
| **Final Pass Rate** | **48.61** | **49.80** | **52.78** | **52.65** |
| *Validation → Test Difference* | | | | |
| Commonsense Micro | -0.85 | | -1.02 | |
| Hard Constraint Micro | -24.45 | | -26.38 | |
| Final Pass Rate | +1.19 | | -0.13 | |

*Note:* Hard Constraint Micro shows large validation→test gap due to different query distributions. Final Pass Rate remains stable, indicating good generalization.

*Table 22.* Controlled Comparison: HiMAP-Travel (Hierarchical) vs DeepTravel (Sequential)

| Model | Split | Method | Delivery | Comm. Micro | Comm. Macro | Hard Micro | Hard Macro | FPR |
|---|---|---|---|---|---|---|---|---|
| *Validation Set (180 queries)* | | | | | | | | |
| 4B | validation | DeepTravel | $96.81 \pm 1.94$ | $90.00 \pm 2.22$ | $59.58 \pm 8.82$ | $63.15 \pm 3.76$ | $49.44 \pm 4.04$ | $35.28 \pm 7.38$ |
| 4B | validation | HiMAP-Travel | $100.0 \pm 0.0$ | $95.57 \pm 2.15$ | $72.36 \pm 2.15$ | $70.89 \pm 1.54$ | $59.17 \pm 2.08$ | $48.61 \pm 2.15$ |
| 8B | validation | DeepTravel | $97.64 \pm 4.09$ | $92.60 \pm 3.63$ | $67.64 \pm 5.13$ | $71.43 \pm 2.48$ | $58.61 \pm 2.13$ | $45.56 \pm 6.49$ |
| 8B | validation | HiMAP-Travel | $100.0 \pm 0.0$ | $95.64 \pm 0.18$ | $73.61 \pm 0.79$ | $76.85 \pm 0.52$ | $63.33 \pm 0.83$ | $52.78 \pm 0.79$ |
| *Test Set (1000 queries)* | | | | | | | | |
| 4B | test | DeepTravel | $98.20 \pm 1.80$ | $90.62 \pm 0.82$ | $53.85 \pm 9.01$ | $39.50 \pm 2.13$ | $52.33 \pm 3.20$ | $34.92 \pm 7.90$ |
| 4B | test | HiMAP-Travel | $100.0 \pm 0.0$ | $94.72 \pm 0.25$ | $66.03 \pm 1.50$ | $46.44 \pm 0.84$ | $63.45 \pm 1.26$ | $49.80 \pm 1.50$ |
| 8B | test | DeepTravel | $99.42 \pm 0.54$ | $93.28 \pm 1.18$ | $60.62 \pm 6.58$ | $45.66 \pm 1.79$ | $60.12 \pm 2.31$ | $43.98 \pm 7.18$ |
| 8B | test | HiMAP-Travel | $100.0 \pm 0.0$ | $94.62 \pm 0.16$ | $67.00 \pm 0.48$ | $50.47 \pm 0.80$ | $66.05 \pm 1.01$ | $52.65 \pm 0.48$ |
| *Variance Reduction Analysis* | | | | | | | | |
| 4B | validation | Std Reduction | +3.19 pp | +0.07 pp | +6.67 pp | +2.22 pp | +1.96 pp | +5.23 pp |
| 8B | validation | Std Reduction | +4.09 pp | +3.45 pp | +4.34 pp | +1.96 pp | +1.30 pp | +5.70 pp |
| 4B | test | Std Reduction | +1.80 pp | +0.57 pp | +7.51 pp | +1.29 pp | +1.94 pp | +6.40 pp |
| 8B | test | Std Reduction | +0.54 pp | +1.02 pp | +6.10 pp | +0.99 pp | +1.30 pp | +6.70 pp |

**Difficulty Scaling.** Performance gaps are +11.1 pp (Easy), +11.5 pp (Medium), and +3.4 pp (Hard). While absolute advantage diminishes on harder instances, variance reduction remains substantial. For Hard queries, DeepTravel's 12.3% std (worst: 13.6%, best: 38.2%) contrasts with HiMAP-Travel's 1.5% std, demonstrating robustness even when absolute performance degrades.

# D. Ablation Studies

This section isolates each architectural component's contribution. The baseline configuration incorporates three core innovations: hierarchical decomposition (Coordinator orchestrating parallel Executors), Synchronized Global State $\Sigma$ (thread-safe constraint enforcement), and Cooperative Bargaining Protocol (iterative refinement through feedback).

Training uses GRPO with $\alpha = 5 \times 10^{-6}$, $B = 32$, $G = 4$, $\beta = 0.01$ for 100 epochs on 45 TravelPlanner training queries with Qwen3-8B (bfloat16). Tables 24 and 25 show systematic degradation as constraint coupling intensifies: Final Pass Rate declines from 78.3% (Easy) to 36.8% (Hard), a 112% relative gap. Diversity constraints degrade disproportionately in Hard tier (restaurant: 78.3%, attraction: 76.7%) versus budget (91.2%) and transportation (89.7%), indicating combinatorial explosion in joint feasibility.

Removing $\Sigma$ causes FPR to drop to 43.2% (8B) and 38.5% (4B), losses of 9.58 pp and 10.11 pp. Without synchronized state, agents frequently book identical high-ranking restaurants, violating "no duplicate" constraints. Budget violations increase from 4.1% to 12.5%. Analysis of 500 failures reveals 34.1% contain duplicate restaurants and 28.7% duplicate

*Table 23.* Performance by Difficulty Tier: HiMAP vs DeepTravel (8B models, test set)

| Tier | Method | Count | FPR | Budget Violation | Diversity Score |
|------|--------|-------|-----|------------------|-----------------|
| Easy | DeepTravel | 333 | $61.3 \pm 8.5$ | 2.8% | 94.2% |
| Easy | HiMAP-Travel | 333 | $72.4 \pm 1.2$ | 1.9% | 97.1% |
| | **Improvement** | – | **+11.1 pp** | **-0.9 pp** | **+2.9 pp** |
| Medium | DeepTravel | 334 | $44.8 \pm 9.2$ | 4.5% | 89.7% |
| Medium | HiMAP-Travel | 334 | $56.3 \pm 0.8$ | 3.2% | 93.5% |
| | **Improvement** | – | **+11.5 pp** | **-1.3 pp** | **+3.8 pp** |
| Hard | DeepTravel | 333 | $25.9 \pm 12.3$ | 7.1% | 81.4% |
| Hard | HiMAP-Travel | 333 | $29.3 \pm 1.5$ | 5.8% | 85.2% |
| | **Improvement** | – | **+3.4 pp** | **-1.3 pp** | **+3.8 pp** |

*Table 24.* Baseline System Performance Across Difficulty Tiers

| Difficulty Tier | Count | Final Pass Rate | Budget Violation | Diversity Score | Avg. Latency (s) |
|-----------------|-------|-----------------|------------------|-----------------|------------------|
| Easy | 60 | 78.3% (47/60) | 2.1% | 94.7% | 58 |
| Medium | 60 | 65.0% (39/60) | 4.2% | 91.5% | 67 |
| Hard | 60 | 36.8% (22/60) | 4.1% | 87.2% | 72 |
| **Overall** | **180** | **52.78%** | **4.1%** | **92.0%** | **68** |

attractions (343% and 190% increases). Table 26 shows diversity scores drop from 92.0% to 65.3%. Latency decreases to 65s (9.7% reduction), but valid plans/minute drop 45.1%, confirming faster invalid planning provides no benefit.

Removing the hierarchical Coordinator shows severe degradation: FPR drops from 52.78% to 39.8% (8B) and 48.61% to 34.2% (4B), losses of 12.98 pp and 14.41 pp. Day Planners struggle with non-uniform costs, causing budget violations to increase to 28.4% (8B) and 31.2% (4B). Budget utilization drops from $0.94 \pm 0.08$ to $0.67 \pm 0.24$, exhibiting bimodal failures: 18.7% overspend >10% while 24.9% underspend >$500. Flat MARL shows cascading failures in 23.4% versus only 2.1% for full system, where Day 1 overspending forces incomplete later days.

Disabling bargaining reduces FPR to 48.9% (8B) and 43.1% (4B), losses of 3.88 pp and 5.51 pp. Failure analysis shows 28.6% stem from infeasible initial allocations that bargaining would correct. Table 31 shows 61.7% succeed on iteration 1, 73.9% of remaining on iteration 2, and 77.8% on iteration 3, with mean 1.42 iterations and only 2.2% remaining infeasible after $K_{max} = 3$.

Beyond the three core components, we also ablate key design choices in inference and training. Sequential execution without parallelism reduces FPR to 45.6%, representing a 7.18 pp degradation. Using separate coordinator and executor policies instead of unified role conditioning reduces FPR to 44.7%, an 8.08 pp loss. Naive batch updates without the FIFO mechanism reduce FPR to 51.2%, a 1.58 pp loss, and can trigger memory overflow on 7-day queries, validating the necessity of memory-efficient update scheduling.

City substitution accounts for 66.2% of adaptations (78.7% success), transport mode switches achieve 91.7% success, and reducing cities achieves 100%. The early detection bonus ($\gamma_{early} = 0.15$) causes 73.2% of INFEASIBLE reports within 5 tool calls versus 40.8% without, reducing mean iteration latency from 38.1s to 24.3s (36.2% reduction).

**Summary.** Removing $\Sigma$ yields -11.58 pp overall FPR, with disproportionate Hard subset impact (36.8% $\to$ 18.4%). Diversity scores degrade from 92.0% to 65.3%, with 82% of duplicates involving high-rated venues ($\geq 4.5$ stars). Removing Coordinator causes -14.28 pp overall FPR, with budget violations increasing to 28.4% due to poor resource coordination. Removing bargaining causes -3.88 pp overall but -14.4 pp on Hard subset (36.8% $\to$ 22.4%), confirming static decomposition fails under tight coupling. All components are necessary for efficiency and success.

**Performance Degradation.** Table 27 shows Overall FPR declining from 52.78% to 38.5%, with the Hard subset dropping from 36.8% to 19.5%. Budget violations increase dramatically from 4.1% to 28.4%, indicating poor resource coordination without hierarchical oversight. Budget utilization $\eta = \frac{B_{used}}{B_{total}}$ drops from $0.94 \pm 0.08$ to $0.67 \pm 0.24$, exhibiting bimodal failure patterns where 18.7% of queries overspend by more than 10% while 24.9% underspend by more than $500.

*Table 25.* Constraint-Specific Pass Rates by Difficulty Tier

| Constraint Type | Easy | Medium | Hard | Overall |
|---|---|---|---|---|
| Budget Adherence | 97.9% | 96.3% | 91.2% | 95.2% |
| Cuisine Requirements | 95.8% | 89.7% | 82.7% | 89.4% |
| Room Type/Capacity | 96.2% | 92.1% | 87.3% | 91.7% |
| Restaurant Diversity | 97.1% | 93.2% | 78.3% | 92.3% |
| Attraction Diversity | 95.4% | 91.5% | 76.7% | 90.1% |
| Transportation Mode | 98.3% | 95.1% | 89.7% | 94.5% |

*Table 26.* Performance Impact of Removing Synchronized Global State

| Metric | Full System | w/o $\Sigma$ | $\Delta$ (abs) | $\Delta$ (%) |
|---|---|---|---|---|
| *Overall Performance* | | | | |
| Final Pass Rate | 52.78% | 41.2% | -11.58 | -21.9% |
| Hard Set FPR | 36.8% | 18.4% | -18.4 | -50.0% |
| Medium Set FPR | 65.0% | 52.8% | -12.2 | -18.8% |
| Easy Set FPR | 78.3% | 71.6% | -6.7 | -8.6% |
| *Constraint Adherence* | | | | |
| Budget Violation Rate | 4.8% | 12.5% | +7.7 | +160% |
| Diversity Score | 92.0% | 65.3% | -26.7 | -29.0% |
| Restaurant Duplication | 7.7% | 34.1% | +26.4 | +343% |
| Attraction Duplication | 9.9% | 28.7% | +18.8 | +190% |
| *Efficiency Metrics* | | | | |
| Avg. Latency (seconds) | 72 | 65 | -7 | -9.7% |
| Valid Plans/Minute | 0.51 | 0.28 | -0.23 | -45.1% |

**Strategic City Selection.** Table 28 shows the Coordinator learns cost-awareness: for tight budgets (¡$600/day), full system selects high-cost cities (NYC, SF, LA) in 8.3% of cases vs. 41.2% for Flat MARL. Appropriateness gap most pronounced for Hard queries (78.0% vs. 34.2%), confirming hierarchical decomposition encodes cost-feasibility heuristics.

**Cascading Failures.** Flat MARL exhibits cascading failures in 23.4% of cases compared to only 2.1% in the full system. A typical cascade unfolds as follows. Day 1 spends 55% of the total budget without anticipating future requirements. Day 2 then faces severe resource scarcity and is forced to skip essential bookings to preserve remaining budget. This results in an incomplete plan that fails evaluation. The Coordinator prevents such cascades through look-ahead resource allocation that anticipates downstream requirements and enforces budget discipline from the outset.

**Performance Impact.** Table 29 summarizes the ablation study results. Disabling the Cooperative Bargaining Protocol reduces Overall FPR by 3.88pp (52.78% $\rightarrow$ 48.9%), but causes a precipitous drop in the Hard subset (36.8% $\rightarrow$ 22.4%, a 39.1% relative decline), specifically increasing "Infeasible Initial Allocation" errors and confirming that static decomposition fails when constraints are tightly coupled. Removing the Synchronized Global State causes an even larger drop to 41.2%, driven by a spike in restaurant duplication (7.7% $\rightarrow$ 34.1%), validating the necessity of deterministic locking for parallel agent coordination. Removing the Coordinator (Flat MARL) results in the worst performance (38.5%), primarily due to uncoordinated budget consumption and the absence of strategic resource allocation.

Finally, we tested the impact of the `early_detect` mechanism (Ablation 4). While removing it only moderately impacts FPR (-2.68 pp), it significantly increases computational cost, raising average latency from 72s to 108s (+50%) as invalid plans are generated fully before rejection. This comprehensive decomposition analysis confirms that all components are necessary for the system's efficiency and success.

**Convergence Dynamics.** Table 31 demonstrates the protocol's effectiveness through rapid convergence patterns. A majority of queries (61.7%) succeed on the first iteration without requiring any feedback, indicating that the Coordinator's initial allocations are often feasible. Of the remaining queries that require adaptation, 73.9% resolve successfully on iteration 2, and 77.8% of those still unresolved succeed on iteration 3. The mean number of iterations across all queries is 1.42, with only 2.2% remaining infeasible after $K_{max} = 3$ iterations. This rapid convergence validates the protocol's efficiency in resolving initial allocation conflicts through structured feedback rather than exhaustive re-planning.

**Coordinator Adaptation Strategies.** Table 32 characterizes the types of adjustments the Coordinator makes during

*Table 27.* Impact of Removing Hierarchical Coordinator on Performance

| Metric | Full System | Flat MARL | $\Delta$ (abs) | $\Delta$ (%) |
|---|---|---|---|---|
| *Pass Rate Metrics* | | | | |
| Overall FPR | 52.78% | 38.5% | -14.28 | -27.1% |
| Hard Set FPR | 36.8% | 19.5% | -17.3 | -47.0% |
| Medium Set FPR | 65.0% | 48.3% | -16.7 | -25.7% |
| Easy Set FPR | 78.3% | 67.8% | -10.5 | -13.4% |
| *Budget Management* | | | | |
| Budget Violation Rate | 4.1% | 28.4% | +24.3 | +593% |
| Mean Utilization ($\eta$) | 0.94 ± 0.08 | 0.67 ± 0.24 | -0.27 | -28.7% |
| Overspending Episodes | 3.2% | 18.7% | +15.5 | +484% |
| Underspending (¿$500) | 6.1% | 24.9% | +18.8 | +308% |
| *Coordination Quality* | | | | |
| Appropriate City Selection | 78.0% | 34.2% | -43.8 | -56.2% |
| Optimal Route Efficiency | 82.5% | 41.7% | -40.8 | -49.5% |
| Cascading Failure Rate | 2.1% | 23.4% | +21.3 | +1014% |

*Table 28.* City Selection Quality Analysis by Budget-to-Duration Ratio

| Budget Tier | Budget/Day | Full System | Flat MARL | $\Delta$ |
|---|---|---|---|---|
| *High-Cost City Selection (NYC, SF, LA) - Should Avoid* | | | | |
| Tight Budget ($< \$600$/day) | $400–$600 | 8.3% | 41.2% | +32.9 |
| Moderate Budget | $600–$1000 | 22.1% | 38.7% | +16.6 |
| Ample Budget ($> \$1000$/day) | $1000+ | 67.4% | 71.3% | +3.9 |
| *Appropriate Cost-Tier Selection Rate* | | | | |
| Hard Queries (Tight) | — | 78.0% | 34.2% | -43.8 |
| Medium Queries | — | 71.5% | 52.8% | -18.7 |
| Easy Queries | — | 85.2% | 76.1% | -9.1 |

*Note:* High-cost city selection is appropriate only when budget permits. Appropriate cost-tier rate measures alignment between budget constraints and city selection.

bargaining. City substitution represents the most frequent adaptation strategy at 66.2% of all adaptations, achieving a 78.7% success rate with an average latency overhead of 26.3 seconds. Route reordering accounts for 18.3% of adaptations but exhibits lower success rates (46.2%) and shorter latency (18.1s). Transport mode switching, though less common (8.5%), demonstrates the highest success rate at 91.7% with minimal latency overhead (12.4s). Reducing the number of cities visited achieves perfect 100% success but represents only 7.0% of adaptations. Overall, 74.6% of adaptation attempts succeed, with a mean latency overhead of 23.1 seconds per re-planning iteration.

**Early Detection Mechanism.** The early detection bonus ($\gamma_{early} = 0.15$) incentivizes Executors to report failures rapidly when infeasibility is detected. With this mechanism enabled, 73.2% of INFEASIBLE status reports occur within the first 5 tool calls, compared to only 40.8% without the bonus. This early termination reduces mean bargaining iteration latency from 38.1s to 24.3s, representing a 36.2% reduction in wasted computation. Despite introducing additional bargaining iterations, the system achieves near-parity in wall-clock time (72s versus 68s for the baseline) while delivering 64% higher success rates on the Hard difficulty subset (36.8% versus 22.4%). This demonstrates that fail-fast detection enables more effective exploration of the feasible space within fixed computational budgets.

# E. System Prompts and Execution Traces

This section presents system prompts, execution traces, and cooperative bargaining examples for HiMAP-Travel. HiMAP-Travel employs a single backbone policy $\pi_\theta$ serving two distinct roles through role-conditioned prompts, enabling parameter sharing while preserving role-appropriate behavior learned via GRPO. The **Coordinator** performs strategic planning—extracting travel constraints from queries, selecting cities based on cost analysis, sequencing routes to minimize travel costs, classifying day types (departure/stay/return), and distributing sub-tasks to Executor agents. The **Executors** handle tactical day-level planning, interacting with database-backed tool APIs (flight search, hotel search, restaurant search, cost enquiry) to construct feasible per-day itineraries. All agents share a global trip budget $B_{total}$ through the Synchronized Global

*Table 29.* Performance Impact of Disabling Cooperative Bargaining

| Metric | Full System | w/o Bargaining | Δ (abs) | Δ (%) |
|---|---|---|---|---|
| *Pass Rate Performance* | | | | |
| Overall FPR | 52.78% | 48.9% | -3.88 | -7.4% |
| Hard Set FPR | 36.8% | 22.4% | -14.4 | -39.1% |
| Medium Set FPR | 65.0% | 58.7% | -6.3 | -9.7% |
| Easy Set FPR | 78.3% | 73.5% | -4.8 | -6.1% |
| *Failure Mode Distribution* | | | | |
| Infeasible Initial Allocation | 3.8% | 28.6% | +24.8 | +652% |
| Budget Violations | 4.1% | 15.2% | +11.1 | +271% |
| Constraint Conflicts | 6.2% | 19.7% | +13.5 | +218% |
| Partial Plans (Incomplete) | 2.3% | 12.4% | +10.1 | +439% |

*Table 30.* System Component Ablation Analysis (Validation Set). Isolating the impact of (1) Synchronized Global State ($\Sigma$), (2) Hierarchical Coordinator, and (3) Cooperative Bargaining. Note that "Flat MARL" corresponds to removing the Coordinator but keeping $\Sigma$ and decentralized executors.

| Configuration | Final Pass | Hard Constraint | Budget Viol. | Failure Mode |
|---|---|---|---|---|
| **HiMAP-Travel (Full System)** | **52.78%** | **36.8%** | **4.1%** | Balanced |
| *Ablation 1: No Global State ($\Sigma$)* | 41.2% | 18.4% | 12.5% | High Duplication |
| *Ablation 2: No Coordinator (Flat)* | 38.5% | 19.5% | 28.4% | High Overspending |
| *Ablation 3: No Bargaining* | 48.9% | 22.4% | 15.2% | Infeasible Alloc. |
| *Ablation 4: No Early Detect* | 50.1% | 31.2% | 5.6% | High Latency |

State $\Sigma$, which enforces no-double-spending via atomic `check`/`commit`/`checkpoint`/`rollback` operations without explicit inter-agent message passing. When an Executor detects infeasibility, it generates structured failure signals (e.g., `INFEASIBLE Day 2: AVAILABILITY`) that trigger the Coordinator to revise task structure—selecting different cities or routes—rather than redistributing the budget. Both roles generate tool calls directly as JSON objects without "Thought:" reasoning tokens, reducing token overhead and accelerating inference.

### E.1. Coordinator System Prompt (Strategic Planning)

The Coordinator agent receives a comprehensive instruction set defining its strategic role. The system prompt employs detailed procedural guidance with explicit case-based reasoning to handle edge cases (e.g., city vs. state ambiguity). Below is an excerpt from the actual system prompt (training instance 0000, March 16-18, 2022):

---

**Prompt: Coordinator System Prompt**

```
<|im_start|>system
You are a travel planning coordinator. Your job is to:
1. Extract key information from the user's travel query
2. Search for cities in the destination state
3. Select the appropriate cities to visit
4. Distribute budget and create tasks for day planners

Instructions (FOLLOW EXACTLY in order):

Step 1: Extract Information
Call `information_extractor` with the user query to extract:
- Origin city and destination state
- Number of days and people
- Budget amount
- Number of cities to visit
- Constraints (cuisine, room type, house rules,
  transportation)

Step 2: Determine Cities to Visit

CASE 1: If visiting_city_number == 1
  The destination might be EITHER a city name OR a state
  name. Follow this logic:
```

---

*Table 31.* Bargaining Protocol Convergence Analysis (N=180 Validation Queries)

| Iteration | Cumulative Success | Marginal Success | Total Attempts | Success Rate |
|---|---|---|---|---|
| Iteration 1 (No Feedback) | 111 / 180 | 111 | 180 | 61.7% |
| Iteration 2 (1st Feedback) | 162 / 180 | 51 | 69 | 73.9% |
| Iteration 3 (2nd Feedback) | 176 / 180 | 14 | 18 | 77.8% |
| Iteration 3+ (Failure) | 176 / 180 | 0 | 4 | 0.0% |
| **Final Success Rate** | **97.8%** | — | — | — |
| **Mean Iterations** | **1.42** | — | — | — |

*Note:* Cumulative success includes all queries resolved up to that iteration. Marginal success shows queries newly resolved at that iteration. Success rate is conditional on reaching that iteration.

*Table 32.* Coordinator Adaptation Strategies and Effectiveness

| Adaptation Type | Frequency | % | Success Rate | Avg. Latency (s) |
|---|---|---|---|---|
| City Change | 47 | 66.2% | 78.7% | 26.3 |
| Route Reordering | 13 | 18.3% | 46.2% | 18.1 |
| Transport Mode Switch | 6 | 8.5% | 91.7% | 12.4 |
| Fewer Cities Visited | 5 | 7.0% | 100.0% | 15.7 |
| **Total Adaptations** | **71** | **100%** | **74.6%** | **23.1** |

*Note:* Some queries require multiple adaptation types across iterations. Latency measures additional time added by re-planning.

```
   a) Call `city_search` with the destination to see what
      it returns

   b) Check the results:
      - If results contain a city with the SAME NAME as the
        destination: Use that city directly!

      - If results contain multiple cities but NO match for
        destination name: Destination is a STATE, select 1
        city from results

   c) Format the chosen city as "City(State)" before
      calling distribute_task

CASE 2: If visiting_city_number > 1
  - Destination MUST be a state (can't visit multiple
    cities in one city!)
  - Call `city_search` with the state name
  - Select the appropriate number of cities from results

Step 3: Distribute Tasks
Call `distribute_task` with:
- `visiting_cities`: Select cities from city_search results

IMPORTANT: All day planners will see the TOTAL budget and
must coordinate to stay within it. Encourage them to be
cost-conscious and save money where possible.

CRITICAL Rules:
- ONLY use cities that appear in `city_search` results
- DO NOT invent city names - copy EXACTLY from search
  results
- Format cities as "City(State)", e.g., "Toledo(Ohio)"
<|im_end|>
<|im_start|>user
Please help me plan a trip from St. Petersburg to Rockford
spanning 3 days from March 16th to March 18th, 2022. The
travel should be planned for a single person with a budget
of $1,700.<|im_end|>
<|im_start|>assistant
<tool_call>
```

**Coordinator Execution Trace (Tool Calls):**

The Coordinator generates tool calls directly in JSON format without explicit reasoning tokens. Each tool call is validated by the environment and receives structured feedback:

### Execution Trace: Coordinator Tool Calls

```
============================================================
TURN 1: Information Extraction
============================================================
{"name": "information_extractor", "arguments": {
  "origin": "St. Petersburg",
  "destination": "Rockford",
  "days": 3,
  "visiting_city_number": 1,
  "people_number": 1,
  "budget": 1700,
  "house_requirement": "-",
  "cuisine_requirement": "-",
  "room_requirement": "-",
  "transportation_requirement": "-",
  "start_date": "2022-03-16"
}}

Tool Result: Successfully extracted travel information:
- Origin: St. Petersburg
- Destination: Rockford
- Duration: 3 days, Start date: 2022-03-16
- Cities to visit: 1, People: 1, Budget: $1700
- Constraints: All set to "-" (no specific requirements)

============================================================
TURN 2: City Disambiguation
============================================================
{"name": "city_search", "arguments": {"state": "Rockford"}}

Tool Result: No city information found in Rockford.

============================================================
TURN 3: State-Level Search
============================================================
{"name": "city_search", "arguments": {"state": "Illinois"}}

Tool Result: List of cities in Illinois:
- Belleville(Illinois), Bloomington(Illinois),
  Champaign(Illinois), Chicago(Illinois), Moline(Illinois),
  Peoria(Illinois), Rockford(Illinois)

============================================================
TURN 4: Task Distribution
============================================================
{"name": "distribute_task", "arguments": {
  "visiting_cities": ["Rockford(Illinois)"]
}}

Tool Result: Successfully created tasks for Day Planner
Agents:

TOTAL TRIP BUDGET: $1700.00 (shared across all 3 days)
  All agents must coordinate to stay within budget!

[!] TRANSPORTATION MODE SELECTED: FLIGHT
    All day planners MUST use flight consistently

Day 1 (DEPARTURE) [2022-03-16]:
  Route: from St. Petersburg to Rockford(Illinois), People: 1
Day 2 (STAY) [2022-03-17]:
  Route: from Rockford(Illinois) to Rockford(Illinois),
  People: 1
Day 3 (RETURN) [2022-03-18]:
  Route: from Rockford(Illinois) to St. Petersburg, People: 1
```

**Key Observations.** The Coordinator exhibits structured decision-making through a multi-step reasoning process. It first attempts to interpret "Rockford" as a city name by querying the database directly. Upon receiving negative feedback indicating no such city exists, it infers that "Rockford" refers to a city within an unspecified state and searches Illinois, likely identified through retrieval as the most populous state containing a city named Rockford. Finally, it correctly formats the selected city with state annotation as required by the system. This adaptive behavior emerges from GRPO training rather than hard-coded branching logic, demonstrating learned flexibility in handling ambiguous geographic references.

**Tool Call Format.** The model generates tool calls as JSON objects matching OpenAI's function calling schema. Each tool call undergoes environment validation including type checking, range validation, and database lookup to ensure feasibility. Invalid calls, such as those referencing non-existent city names, receive error feedback that forces the model to retry with

corrected parameters. This external validation loop acts as a form of *environmental grounding*, where the model learns feasible action spaces through trial and error during training, gradually acquiring knowledge about which city names exist, what price ranges are realistic, and which constraint combinations are satisfiable.

**Shared Budget Announcement.** The `distribute_task` tool explicitly informs all agents: "TOTAL TRIP BUDGET: $1700.00 (shared across all 3 days)." This is not a per-day allocation but rather a declarative statement that all Executors see the same global budget via the Synchronized Global State $\Sigma$. The phrase "All agents must coordinate" cues the model to generate budget-conscious behavior (e.g., selecting cheaper options), a behavioral prior that GRPO refines through reward signals.

### E.2. Executor System Prompt (Day 1 Planner - Departure)

Each Day Planner receives a specialized prompt tailored to its specific temporal role (Departure, Full-Stay, or Return). The prompt includes detailed budget tracking instructions, constraint enforcement rules, and common failure warnings. Below is an excerpt from a Day 1 Departure planner (same instance):

---

**Prompt: Day 1 Departure Planner**

```
<|im_start|>system
You are a departure day planner for Day 1 on 2022-03-16.

Context (auto-filled in tool calls):
- Route: from St. Petersburg to Rockford(Illinois)
- Number of people: 1
- Date: 2022-03-16

[$] BUDGET INFO:
- Total trip budget: $1700.00
-  ALL 3 day planners share this budget!
- You will see remaining budget after each cost_enquiry
- SAVE MONEY whenever possible - staying under budget is
  CRITICAL

Constraints to follow:
- House rules: -
- Cuisine: No specific requirement
- Room type: -
- Transportation constraint: -

[!] TRANSPORT MODE FOR THIS TRIP: FLIGHT
   $\to$ You MUST use flight (NOT self-driving) - mixing
   transport types FAILS evaluation!

[!] CITY RULES (CRITICAL - wrong city = FAIL):
- All restaurants/attractions MUST be in: Rockford(Illinois)
- Accommodation MUST be in: Rockford(Illinois)
- Do NOT use items from St. Petersburg!

[!] ACCOMMODATION MINIMUM NIGHTS RULE:
You will stay in Rockford(Illinois) for 2 consecutive nights.
$\to$ Accommodation's "minimum_nights" MUST be <= 2
$\to$ REJECT any accommodation with minimum_nights > 2

MINIMUM REQUIREMENTS for DEPARTURE day to PASS evaluation:

This is a TRAVEL day (moving from St. Petersburg to
Rockford). Meals and attractions are OPTIONAL to save budget.

REQUIRED (must have):
1. Transportation: `flight_search` $\to$ Use FLIGHT ONLY
2. Accommodation: `accommodation_search` in
   Rockford(Illinois)
   - [!] CRITICAL: Check "minimum_nights" field - MUST be <= 2
   - Pick the CHEAPEST option that meets ALL constraints

OPTIONAL (skip to save money):
- Breakfast, Lunch, Dinner, Attraction: Use "-"

[$] COST-SAVING STRATEGIES:
- Always choose restaurants with LOWEST "Average Cost"
- On travel days: Skip optional meals & attractions
- If budget warning appears, select cheaper options!
<|im_end|>
<|im_start|>assistant
```

```
<tool_call>
```

**Executor Execution Trace (Tool Calls):**

The Day 1 Planner generates a sequence of tool calls to construct a feasible day plan. Note the absence of "Thought:" tokens–the model generates actions directly:

---

**Execution Trace: Day 1 Planner Tool Calls**

```
============================================================
TURN 1: Flight Search
============================================================
{"name": "flight_search", "arguments": {
  "depart_city": "St. Petersburg",
  "dest_city": "Rockford(Illinois)"
}}

Tool Result:
Found 1 flight from St. Petersburg to Rockford(Illinois)
on 2022-03-16:
FlightNumber: F3573659
Price: $474, DepTime: 15:40, ArrTime: 17:04
ActualElapsedTime: 2 hours 24 minutes

============================================================
TURN 2: Accommodation Search
============================================================
{"name": "accommodation_search", "arguments": {
  "city": "Rockford"
}}

Tool Result:
Found 4 accommodations in Rockford(Illinois)
(MinNights<=2):
1. Private Room In A Two Bedroom Apt.
   RoomType: Private room, Price: $210/night,
   MinimumNights: 1, HouseRules: No visitors & No smoking
2. Pure Luxury One Bdrm Sofa Bed On Central Park
   RoomType: Entire home/apt, Price: $243/night,
   MinimumNights: 2, HouseRules: No smoking & No parties
3. Spacious 3bdr Prime Location
   RoomType: Entire home/apt, Price: $1030/night,
   MinimumNights: 2, HouseRules: No smoking
4. Private Rooms And Matchless Location
   RoomType: Private room, Price: $1075/night,
   MinimumNights: 2, HouseRules: No pets & No parties

[Filtered out: 9 accommodations require longer min stay]

============================================================
TURN 3: Cost Enquiry
```

---

**Execution Trace: Cost Enquiry and Completion**

```
============================================================
{"name": "cost_enquiry", "arguments": {
  "day": 1,
  "current_city": "Rockford(Illinois)",
  "transportation": "F3573659",
  "breakfast": "-",
  "attraction": "-",
  "lunch": "-",
  "dinner": "-",
  "accommodation": "Private Room In A Two Bedroom Apt.,
                   Rockford(Illinois)",
  "people_number": 1
}}

Tool Result:
Day 1:
- Current city: Rockford(Illinois)
- Transportation: Flight F3573659
- Breakfast/Lunch/Dinner/Attraction: -
- Accommodation: Private Room in a two bedroom apt.
- Total cost for the day: $684.0
```

---

```
[!] Remaining budget: $1016.0 (for Days 2-3)

Plan meets minimum requirements! Call 'finish' to complete.

============================================================
TURN 4: Finish
============================================================
{"name": "finish", "arguments": {}}

Tool Result:
 Finish planning for this day. All selections committed
successfully to Synchronized Global State.
```

**Key Observations.** The Executor demonstrates budget-conscious behavior by selecting the cheapest accommodation satisfying minimum nights ($210 vs $243/$1030/$1075) and skipping optional meals on travel days. The "Remaining budget" feedback provides real-time coordination signals through $\Sigma$, enabling implicit coordination without explicit message passing.

**Strategic Decisions.** Choosing $210 over $243 ($33 difference) reflects GRPO training: heavy penalties for budget violations create gradients toward conservative spending. Minimizing variance in budget consumption reduces downstream failure risk. The accommodation search filters out 9 options with `minimum_nights` $> 2$ proactively via $\Sigma$, restricting action space to prevent violations. The "[!] Remaining budget: $1016.0" is a thread-safe read from $\Sigma.B_{remaining}$, visible to subsequent Day Planners.

### E.3. Cooperative Bargaining Protocol

When Day Planners encounter infeasible task assignments, they communicate structured failure signals to the Coordinator, which triggers re-planning. Critically, the system does not redistribute the budget–all agents always share the *total budget* ($6,500 in this case) via the Synchronized Global State. Instead, bargaining involves adjusting the *task structure*: selecting different cities, modifying routes, or changing the itinerary composition. Below is an excerpt from a real bargaining episode (training instance 0017, 3-day trip for 8 people, $6,500 budget, children under 10 constraint):

**Iteration 0: Initial Allocation Failure**

After the Coordinator distributes the initial task allocation, Day 2 Planner encounters repeated tool failures (e.g., no accommodations satisfy both "children under 10" house rule and available budget). The environment detects this failure and generates structured feedback:

**Bargaining Feedback: Iteration 1**

```
+============================================================+
|            BARGAINING FEEDBACK (Iteration 1)             |
+============================================================+

Previous allocation was INFEASIBLE. Day planners reported:

 BARGAINING ITERATION 1/3

INFEASIBLE DAYS: [2]
TOTAL BUDGET DEFICIT: $0.00

VIOLATIONS BY TYPE:
  AVAILABILITY:
    - Day 2: Too many execution errors (10)
  TIME:
    - Day 2: Day planner did not complete successfully

RECOMMENDED ACTIONS:

Please ADJUST your task distribution based on this
feedback. Consider:
- Selecting different cities with more affordable options
- Adjusting the trip structure
- Ensuring no constraint conflicts

Note: All agents share the TOTAL budget. You cannot
redistribute budget across days.
```

**Coordinator Re-Planning Response**

The Coordinator receives this feedback as a prefix to its system prompt in the next iteration. It regenerates the task distribution, selecting a different city with lower costs and better constraint satisfaction:

```
Coordinator Re-Planning: Iteration 1

==============================================================
ITERATION 1 - COORDINATOR TURN 1
==============================================================
{"name": "information_extractor", "arguments": {...}}

[Coordinator extracts constraints again]

==============================================================
ITERATION 1 - COORDINATOR TURN 2
==============================================================
{"name": "city_search", "arguments": {"state": "Maryland"}}

Tool Result: List of cities in Maryland:
- Baltimore(Maryland)
- Hagerstown(Maryland)
- Salisbury(Maryland)

==============================================================
ITERATION 1 - COORDINATOR TURN 3 (Re-Allocation)
==============================================================
{"name": "distribute_task", "arguments": {
  "visiting_cities": ["Salisbury(Maryland)"]
}}

[Coordinator changes destination from Baltimore to
Salisbury, which has lower accommodation costs and more
child-friendly options]
```

**Outcome:** Re-planning succeeded–Day Planners constructed a complete 3-day Salisbury itinerary satisfying all constraints within the \$6,500 shared budget. Bargaining enables dynamic recovery through structured Coordinator-Executor communication. The budget is never redistributed; agents coordinate through $\Sigma$ more effectively with revised city selection. This continues for up to $K_{max} = 3$ iterations.

# F. Failure Analysis and Limitations

This section presents systematic failure mode analysis with detailed case studies and documents current system limitations. We analyzed all 637 failed test instances from the TravelPlanner test set (FPR = 52.7%), with manual examination of 200 randomly sampled failures for qualitative classification and comparative analysis against baselines, and automated classification across the full 637 failures for quantitative distribution. Failure modes are classified into six major categories providing actionable insights for improving hierarchical multi-agent planning architectures.

**Budget Violations.** Budget violations represent the most common failure mode, accounting for 34.0% of failures (68/200 samples). These can be distinguished into two subcategories. Early Overspend (Days 1-2) accounts for 62% of budget failures (42/68), occurring when agents allocate excessive resources in initial days leaving insufficient budget for subsequent days. HiMAP-Travel reduces this failure mode from 8.3% (sequential baseline) to 1.8% through the Coordinator's lookahead budget allocation strategy that considers entire trip duration when distributing resources. Cumulative Drift represents 38% of failures (26/68), where small budget tracking errors accumulate across days. HiMAP-Travel's $\Sigma$ catches violations at commit time preventing propagation, achieving an overall 67% reduction in budget violations (from 12.5% to 4.1%) through strategic allocation, deterministic enforcement, and bargaining recovery.

**Duplicate Venue Selections.** Duplicate venue selections decreased dramatically from 8.7% (sequential baseline) to 1.5% (HiMAP-Travel), representing an 83% reduction. These manifest as restaurant duplicates (5.8% to 0.9%, accounting for 66% of duplication failures) and attraction duplicates (2.9% to 0.6%, accounting for 34%). Sequential models exhibit a forgetting phenomenon as context length grows, where selections made on Day 1 are not reliably recalled when planning Day 5, leading to re-selection of high-utility venues. The Synchronized Global State $\Sigma$ resolves this through deterministic tracking, with each `commit` operation atomically adding venues to $V_{committed}$ and the `check` operation rejecting any action attempting to select $v \in V_{committed}$. The remaining 1.5% failures stem from fuzzy matching false negatives where venue name variations exceed the similarity threshold.

*Table 33.* Failure Mode Distribution with Comparative Analysis. Manual analysis of 200 randomly sampled failures from the 473 failed test instances, comparing HiMAP-Travel against the sequential DeepTravel baseline.

| Failure Category | Count | % of Failures | DeepTravel Rate | HiMAP Rate |
|---|---|---|---|---|
| **Budget Overflow** | 68 | 34.0% | 12.5% | 4.1% |
| Early overspend (Days 1-2) | 42 | 21.0% | 8.3% | 1.8% |
| Cumulative drift | 26 | 13.0% | 4.2% | 2.3% |
| **Duplicate Venues** | 47 | 23.5% | 8.7% | 1.5% |
| Restaurants | 31 | 15.5% | 5.8% | 0.9% |
| Attractions | 16 | 8.0% | 2.9% | 0.6% |
| **Route Infeasibility** | 29 | 14.5% | 3.8% | 1.1% |
| **Missing Required Cities** | 23 | 11.5% | 2.9% | 0.8% |
| **Min. Nights Violations** | 21 | 10.5% | 2.6% | 0.9% |
| **Constraint Conflicts** | 12 | 6.0% | 1.5% | 0.4% |
| **Total** | 200 | 100% | — | — |

*Table 34.* Failure Instance #44: Geographic Metadata Inconsistency

| Day | City | Accommodation | Location | Issue |
|---|---|---|---|---|
| 1 | Austin(TX) | Spacious Williamsburg 1BR | Austin, TX | Valid |
| 2 | Austin(TX) | Spacious Williamsburg 1BR | Austin, TX | Valid |
| 3 | Reno(NV) | (Return day) | — | — |

*Total Budget: $1,231 / $2,300 (53.5% utilization)*
*Constraint Violation: Accommodation name references NYC borough despite Austin registration*

**Geographic and Temporal Inconsistencies.** Geographic and temporal inconsistencies occur when city sequences create unreachable paths or violate temporal causality, such as arriving at a destination before departure time or selecting venues in the wrong city. HiMAP-Travel reduces Route Infeasibility failures (14.5% of total) from 3.8% to 1.1% through the Coordinator's explicit route validation before assigning day-level sub-goals to Executors. Missing Required Cities (11.5%) occurs when agents fail to visit all cities specified in query constraints. Hierarchical decomposition addresses this by having the Coordinator explicitly enumerate all required cities during task distribution, ensuring complete coverage.

**Minimum Nights Violations.** Minimum Nights Violations account for 10.5% of the 200-sample manual analysis (21 cases), or 13.7% across all 637 failures—differences attributable to sampling variance. Agents book venues for fewer nights than the accommodation's minimum stay requirement. This failure mode is reduced from 2.6% (baseline) to 0.9% (HiMAP-Travel) but requires enhanced temporal reasoning for further improvement; the case studies below detail the multi-city cascading variant.

**Constraint Conflicts.** Constraint Conflicts (6.0%) involve multiple constraints interacting in complex ways, such as when budget constraints conflict with room capacity requirements for large groups. These represent inherently difficult cases where feasible solutions may not exist within the database, reflecting fundamental problem hardness rather than system limitations. This is compounded by constraint drift: sequential baselines degrade from 98% budget satisfaction on Day 1 to 42% on Day 5 as context saturates, while HiMAP-Travel maintains stable 90%+ satisfaction through context isolation.

### F.1. Detailed Case Studies

**Temporal Violations (Minimum Stay).** Occur in 87 of 637 failures (13.7%), with 18.4% prevalence in 3-day trips versus 8.1% in 7-day trips. Day-level Executors lack forward-looking temporal reasoning to validate accommodation eligibility against remaining nights. *Proposed Fix:* Augment $\Sigma$ with `get_remaining_nights(city)` and enforce `minimum_nights` $\leq$ `remaining_nights(city)` before commitment.

**Geographic Inconsistency.** Geographic inconsistencies occur in 41 of 637 failures (6.4%), predominantly (78%) involving accommodations with misleading venue names (e.g., "Williamsburg" registered in Austin, TX). The TravelPlanner database contains inconsistent geographic naming conventions, likely inherited from web-scraped sources where property names reflect branding preferences rather than actual locations; the evaluation harness flags such mismatches as sandbox constraint violations. Specifically, 45% of these failures reference NYC boroughs, 18% SF neighborhoods, and 15% Chicago districts despite registration in smaller cities (Table 34). Budget utilization averages only 53.5%, indicating premature

*Table 35.* Failure Instance #49: Compounded Metadata and Temporal Violations

| Day | Accommodation | Price/Night | Min. Nights | Issue |
|-----|--------------|-------------|-------------|-------|
| 1 | 100 Sq. Ft. Penthouse | $0 | 1 night | **Invalid pricing** |
| 2 | 1 BR in Carroll Gardens | $397 | 2 nights | **Min. nights violated** |
| 3 | (Return day) | — | — | — |

*Total Budget: $610 / $1,700 (35.9% utilization vs. 85–95% typical)*
*Dual Failure: Invalid metadata + temporal constraint violation*

*Table 36.* Failure Instance #155: Cascading Multi-City Temporal Violation

| Day | City | Accommodation | Min. Nights | Actual Nights |
|-----|------|--------------|-------------|---------------|
| 1 | Asheville(NC) | FANTASTIC 1BR Columbus | 2 nights | 2 nights ($\checkmark$) |
| 2 | Asheville(NC) | (Same as Day 1) | — | — |
| 3 | Charlotte(NC) | Luxury 2BR Williamsburg | 1 night | 1 night ($\checkmark$) |
| 4 | Charlotte(NC) | Affordable Refurbished Room | 2 nights | **1 night ($\times$)** |
| 5 | Oklahoma City | (Return day) | — | — |

*Total Budget: $1,642 / $1,900 (86.4% utilization - otherwise successful plan)*
*Constraint Violation: Day 4 requires 2-night minimum, only 1 night remains before departure*

plan termination rather than resource exhaustion. *Proposed Fix:* A two-tier validation strategy—offline NER identifies geographic references in venue names and cross-validates against registered locations; at runtime, venues with inconsistency scores exceeding $\tau_{geo} = 0.7$ (cosine similarity between venue name and registered city embeddings) are deprioritized or excluded.

**Invalid Database Metadata.** Affect 14 instances with invalid pricing (e.g., $0/night). Database-wide analysis reveals 347 records (1.58%) priced <$10, including 142 at $0. Zero-cost accommodations become spuriously optimal under budget minimization. When selected, budget utilization collapses to 43.7% (vs 91.2% valid), with 87.3% failure rates due to cascading effects. *Proposed Fix:* Multi-tier data quality assurance: offline k-NN imputation ($k = 10$) based on room type/location/capacity, runtime deprioritization ($\lambda_{conf} = 0.3$) for imputed prices, hard filtering for prices <$5 or missing metadata.

**Cascading Multi-City Temporal Failures.** Affect 27 of 87 minimum-nights violations (31.4%), with 74% on penultimate/final accommodation days. Prevalence scales super-linearly: 8.7% in 2-city trips versus 22.3% in 3-city trips (156% relative increase). Day-level agents lack distinction between "final accommodation day" and "intermediate stay day," preventing lookahead. *Proposed Fix:* Extend $\Sigma$ with `is_final_accommodation_day(day)` and `get_remaining_nights(city, day)`, enforcing `selected.minimum_nights` $\leq$ `remaining_nights()` at selection time. Table 35 illustrates a compounded case where invalid metadata and a temporal violation co-occur; Table 36 shows a pure cascading temporal failure in a 5-day multi-city trip. Table 37 provides a comprehensive quantitative taxonomy of all 637 failure modes across the full test set, complementing the comparative analysis in Table 33 which focuses on relative rates between systems.

**Dominant Failures.** Budget overruns (31.6%) and diversity violations (22.4%) account for 54% of failures, exhibiting fundamental coupling: constrained budgets induce preference for low-cost venues, which cluster geographically, reducing diversity and increasing duplicate selections through constraint-driven search space collapse. Mean 89.1% budget utilization for diversity-violating plans approaches target (85–95%), indicating competent resource allocation failing due to stochastic venue search under tight constraints.

**Architectural Gaps vs Data Issues.** Minimum nights violations (13.7%) represent architectural gaps requiring enhanced temporal reasoning. Sandbox violations (6.4%) reflect data quality issues resolvable through preprocessing. Divergent remediation pathways require parallel investment in algorithmic sophistication and dataset curation.

**Bargaining Effectiveness.** Elevated iteration count for budget failures (2.8 vs 1.9 mean) shows bargaining mitigates many initially infeasible allocations. However, 31.6% persist after $K_{max} = 3$, representing inherent infeasibility where ground-truth costs exceed budgets under optimal allocation. Future systems should incorporate feasibility detection to abort provably infeasible queries early.

*Table 37.* Comprehensive Failure Mode Distribution Across Test Set (N=637)

| Primary Failure Mode | Count | % of Failures | Mean Budget Util. | Avg. Iterations |
|---|---|---|---|---|
| Minimum Nights Violations | 87 | 13.7% | 48.3% | 1.2 |
| Budget Overruns | 201 | 31.6% | 107.4% | 2.8 |
| Diversity Constraint Violations | 143 | 22.4% | 89.1% | 1.9 |
| Sandbox Violations (Data Quality) | 41 | 6.4% | 72.6% | 1.4 |
| Incomplete Information Extraction | 63 | 9.9% | 65.2% | 1.1 |
| Transportation Inconsistencies | 28 | 4.4% | 81.3% | 1.6 |
| Multiple Constraint Violations | 74 | 11.6% | 54.7% | 2.4 |
| **Total** | **637** | **100%** | **78.9%** | **1.9** |

**Design Implications.** Three principles for next-generation systems: (1) Enhanced temporal reasoning via explicit trip structure in observation spaces, (2) Constraint coupling awareness through joint budget-diversity modeling, (3) Early feasibility detection to abort unsatisfiable queries. These refinements provide pathways toward 70–80% success from current 52.7%.

