# OpenReview forum: "HiMAP-Travel: Hierarchical Multi-Agent Planning for Long-Horizon Constrained Travel"
_ICML.cc/2026/Conference — ICML 2026 regular_

### Official Review · Reviewer_DdCr · 2026-03-09

**Soundness:** 2
**Presentation:** 3
**Significance:** 2
**Originality:** 3
**Overall Recommendation:** 3
**Confidence:** 5

**Summary:**

This paper introduces HiMAP-Travel, a hierarchical multi-agent framework designed for long-horizon travel planning. The work focuses on the problem the authors describe as constraint drift, where sequential LLM planners gradually lose track of global constraints such as budget limits or itinerary diversity as the planning context becomes longer and more complex. To address this, the framework separates strategic and tactical planning roles: a Coordinator manages global resource allocation while multiple Day Executors generate daily plans in parallel within isolated contexts.

The system includes several technical components. A synchronized global state mechanism acts as a transactional monitor to enforce constraints such as budget limits and duplicate bookings. The framework also introduces a cooperative bargaining protocol that allows executors to reject infeasible subgoals and trigger replanning rather than relying on later corrections. In addition, the authors train a role-conditioned policy using GRPO, enabling parameter sharing between the coordinator and executor roles, and implement a memory-efficient FIFO update mechanism to control memory growth during long planning traces.

Empirically, the approach improves planning success on the TravelPlanner benchmark, achieving a 52.65% final pass rate with Qwen3-8B, outperforming several recent baselines including ATLAS, MTP, and DeepTravel. The hierarchical design also reduces latency through parallel execution. Additional evaluation on FlexTravelBench shows consistent improvements across multi-turn constraint adaptation scenarios.

**Compliance With Llm Reviewing Policy:**

Affirmed.

**Key Questions For Authors:**

Training data size and generalization:
The model is trained on only 45 queries, which is extremely small for RL-style training. Can the authors provide stronger evidence that the improvements are not due to overfitting to the TravelPlanner distribution, particularly given the large performance drop on the harder constraint subset?

Architecture vs model capability:
The paper reports improvements over ATLAS and MTP using different backbone models. Could the authors provide additional controlled comparisons using the same backbone models to more clearly isolate the architectural contribution of HiMAP-Travel?

Role of the coordinator vs constraint monitor:
The paper introduces both a Coordinator for strategic planning and a Synchronized Global State monitor for enforcing constraints. How much of the final improvement comes from the hierarchical planning structure versus the deterministic constraint monitor?

Parallelism vs context isolation:
The paper claims that context isolation prevents constraint drift. However, the ablation removing parallelism still changes both concurrency and planning structure. Could the authors evaluate a sequential hierarchical version to isolate the effect of context isolation alone?

**Limitations:**

Benchmark scope of TravelPlanner's training split (45 queries) and relatively constrained domain (US domestic travel with fixed database) may not reflect the full complexity of real world planning. The benchmark's hard constraint definition (13 specific rules) is rigid and
hand-crafted, it is unclear how the system would perform under naturalistically specified constraints or open-domain queries.
Scalability of the coordinator must enumerate all required cities and assign day roles before any execution begins. For very long horizons (14+ day trips) or complex multi-destination itineraries, the coordinator’s context would grow substantially, potentially reintroducing the
Constraint Drift problem at the strategic level. This is not analyzed.

The transactional model's real-world validity. The paper notes that rollback "assumes tentative actions can be released cost-free," which is a significant assumption. In real travel booking systems, holds are not always free (some hotel bookings require non-refundable deposits), and the paper's simulation may overestimate real-world applicability of the bargaining protocol.

Remaining failure modes are substantial. At 52.65% FPR, nearly half of test queries fail. The authors identify budget overruns (31.6% of failures), diversity violations (22.4%), and minimum nights violations (13.7%) as dominant failure categories, with proposed fixes that
are not implemented. The gap to hypothetical 70-80% performance requires architectural enhancements not demonstrated in this work.
Single-domain evaluation, the generalization claims (software engineering, supply chain optimization, scientific experiment planning) are speculative and unsupported by any empirical evidence in the paper.

**Strengths And Weaknesses:**

The paper has several clear strengths. The most convincing aspect is the controlled comparison with the DeepTravel baseline. By keeping the backbone model, training method, data, tools, and decoding settings fixed, the authors isolate the architectural effect of HiMAP-Travel. The improvement in final pass rate together with the large reduction in variance suggests the hierarchical design provides a real stability advantage.

The problem formulation is also well motivated. The paper identifies constraint drift as a concrete failure mode in long-horizon planning and demonstrates it empirically: sequential planners gradually lose track of constraints, while the proposed hierarchical design maintains much higher constraint satisfaction through context isolation.

Another strong point is the system design for constraint enforcement. The synchronized global state acting as a transactional monitor is a sensible mechanism for enforcing invariants such as budget limits and preventing duplicate bookings. The ablation results showing performance drops when this component is removed support its importance.

The ablation study is relatively thorough, examining the impact of removing the coordinator, the global state monitor, the bargaining mechanism, and parallelism. This provides useful insight into which components contribute most to performance. The efficiency gains from parallel planning are also credible, with clear evidence of speed improvements for longer planning horizons.

However, several concerns remain. The training dataset is extremely small (45 queries), which raises questions about robustness and generalization. Although trajectory augmentation is used, the large drop between validation and test performance on the harder constraint subset suggests possible overfitting.

There are also fairness issues in the baseline comparisons. Some headline comparisons involve much stronger proprietary models, making it difficult to attribute improvements purely to the architectural design. The controlled DeepTravel comparison helps, but the broader claims should be presented more cautiously.

In addition, some parts of the system are not fully analysed. The interaction between the coordinator’s budget hints and the hard constraint enforcement by the global state monitor is unclear, making it difficult to understand how much each component contributes to the final behaviour.

The training procedure may introduce inconsistencies, since the FIFO update mechanism allows asynchronous updates across roles. The paper does not examine whether this could lead to mismatches between the coordinator’s strategy and executor behaviour during training.

Finally, the parallelism ablation does not clearly separate two effects: latency improvement from concurrency and performance improvement from context isolation. A sequential hierarchical variant would help clarify which factor is responsible for the observed gains.

---

> ### Author Rebuttal · Authors · 2026-03-31
>
> We thank Reviewer DdCr for the exceptionally detailed review and address each concern below. Each question (Q) is already covered within its associated weakness (W): Q1→W1, Q2→W2, Q3→W3, and Q4→W5. Additional limitation concerns are labeled L1–L3.
>
> ---
>
> **W1**
>
> A: We do not see clear evidence of overfitting. **The main evaluation metric (FPR) is stable across splits:** changes only +1.19 pp (4B) and −0.13 pp (8B) from validation to test (Table 17); the larger Hard Constraint Micro drop reflects **query-composition shift**, not overfitting (see our response to **Reviewer kcTT W1**), since Hard Constraint Macro actually improves on test (+4.28 pp for 4B, +2.72 pp for 8B).
>
> Controlled comparison supports this: under the same 45 queries and GRPO setup, HiMAP-Travel still outperforms DeepTravel by +8.67 pp with 93% lower variance, indicating gains come from architecture, not data advantage.
>
> Finally, **FlexTravelBench, a multi-turn constraint adaptation** benchmark, shows strong HiMAP-Travel performance (44.34% / 37.42% FPR), confirming transfer beyond the original single-turn task.
>
> ---
>
> **W2**
>
> A: We appreciate this concern. Our main architectural claim rests entirely on the **controlled DeepTravel comparison** (+8.67 pp, 93% variance reduction under identical backbone, training, and tools — see our response to **Reviewer FJUj W3** for full details). Comparisons against ATLAS (Gemini-2.5-Pro) and MTP (GPT-4) are included as published reference points, not as primary evidence. We will add a clear caveat in the revision that these comparisons reflect both architectural and model-capacity differences.
>
> ---
>
> **W3**
>
> A: Our paper separates two components both conceptually and empirically. The Coordinator’s budget hint is a **soft prior** that guides Executor search, while ($\Sigma$) is a **deterministic atomic check** that enforces hard constraints.
>
> Our ablation disentangles them: removing $\Sigma$ causes a −9.58 pp drop and mainly increases duplicate bookings (7.7% → 34.1%), whereas removing the Coordinator causes a −12.98 pp drop and mainly increases budget misallocation (4.1% → 28.4%) and cascading failures (2.1% → 23.4%). Thus, the Coordinator mainly handles **allocation**, while ($\Sigma$) handles **runtime conflict prevention**. We will make this soft-vs-hard decomposition more explicit in the revision.
>
> ---
>
> **W4**
>
> A: FIFO updates do not compromise training:
> 1) GRPO group validity: Updates are **intra-role** (G=4 trajectories per role), so advantages are not contaminated by cross-role mixing.
> 2) Role-specific convergence differences are benign. Coordinators stabilize earlier due to simpler action spaces, forming an effective curriculum for Executors (see line 986). KL penalty (β=0.01) prevents drift.
> 3) Single-policy coupling: Shared parameters with prompt conditioning ensure executor updates also improve coordinator weights, reducing mismatch risk. Empirically, using separate policies reduces FPR by 8.08 pp under otherwise identical settings (see line 670).
> 4) Empirical stability: Training curves (Figure 2) show monotonic convergence on Hard Constraint satisfaction (~70%) and FPR, with 93% lower variance, indicating no mismatch issues.
>
> ---
>
> **W5**
>
> A: The "w/o Parallelism" ablation **preserves context isolation**: executors still run in isolated contexts conditioned on $z_d$ and $\Sigma$. The **7.18 pp drop** (52.78% → 45.6%) reflects loss of parallel execution, not context isolation.
>
> Parallelism adds two benefits beyond latency:
> 1) **Independent search coverage** — executors explore sub-plans under independent seeds, improving high-reward selection;
> 2) **Training-time diversity** — parallel rollouts generate richer, less correlated signals, enhancing GRPO advantage estimates.
>
> **Evidence for context isolation** appears in Table 37: sequential baselines without isolation see budget satisfaction dropping 98% → 42% from Day 1 to Day 5, which HiMAP-Travel avoids even sequentially.
>
> ---
>
> **L1**
>
> A: See our response to **Review KcTT W2**. Coordinator context scales linearly (~3,600 tokens for 14 days), well within the 8,192-token limit; Executors’ context remains fixed. Although untested beyond 7 days, saturation is unlikely, and we will clarify this in the revision.
>
> ---
>
> **L2**
>
> A: Our framework treats planning as recommendation generation, so rollback is a local operation without side effects. For real booking APIs with non-refundable holds, the Cooperative Bargaining Protocol can use price quotes or availability checks, preserving its benefits without cancellation costs. We will clarify this assumption and its practical implications.
>
> ---
>
> **L3**
>
> A: Failure modes: 52.65% FPR dominated by budget (31.6%), diversity (22.4%), minimum nights (13.7%). Targeted extensions like cost-aware ($\Sigma$), cuisine-set tracking, and finalized schedule propagation could raise FPR. Our failure analysis itself is a contribution, prioritizing transparency over premature optimization.

---

### Official Review · Reviewer_FJUj · 2026-03-11

**Soundness:** 3
**Presentation:** 3
**Significance:** 3
**Originality:** 3
**Overall Recommendation:** 4
**Confidence:** 4

**Summary:**

This paper proposes HiMAP-Travel, a hierarchical multi-agent planning framework designed to address long-horizon constrained planning tasks for LLM agents. The system decomposes travel planning into a strategic coordinator and parallel day-level executors, combined with a synchronized global state for enforcing shared constraints and a cooperative bargaining protocol for resolving infeasible sub-goals. Experiments on the TravelPlanner and FlexTravelBench benchmarks show improvements in Final Pass Rate and latency compared to sequential agent baselines.

**Compliance With Llm Reviewing Policy:**

Affirmed.

**Final Justification:**

Thanks for the detailed rebuttal and AC's excellent work, I vote for acceptance as my concerns have been addressed properly.

**Key Questions For Authors:**

Specifically, can the authors demonstrate similar improvements on other constrained long-horizon agent tasks (e.g., software planning, task scheduling, or tool-based reasoning benchmarks)? Such evidence would strengthen the claim that the architecture addresses a broader class of planning problems rather than a domain-specific benchmark.

**Limitations:**

yes

**Strengths And Weaknesses:**

### Strengths:
1. The paper identifies an important failure mode of sequential LLM agents, i.e., constraint drift under long tool traces, and frames it clearly as a limitation of monolithic planning architectures. This problem formulation is intuitive and grounded in empirical observations.
2. The proposed system integrates hierarchical planning, synchronized constraint monitoring, and cooperative bargaining into a unified framework. The design is technically coherent and appears carefully engineered to address constraint violations in long-horizon planning tasks.

### Weaknesses:
1. Evaluation focuses almost exclusively on travel planning benchmarks. While the authors claim the method generalizes to other constrained planning domains, the paper does not empirically validate this claim beyond the travel domain.
2. The synchronized global state and transactional constraint monitor enforce hard constraints deterministically. While effective, this raises the question of how much of the performance improvement comes from architectural design rather than improved agent reasoning capabilities.
3. [Experimental fairness] Although the paper includes a controlled comparison with DeepTravel, some comparisons rely on previously published results under different models (e.g., Gemini or GPT-4). A stronger experimental setup would re-evaluate more baselines under identical conditions.

---

> ### Author Rebuttal · Authors · 2026-03-30
>
> Thank you for the constructive and thoughtful feedback. We appreciate your recognition of the paper’s motivation and technical design, and we respond below to the weaknesses (W) and questions (Q).
>
> ---
>
> **W1: Generalization to other constrained planning domains beyond travel planning.**
>
> A: We address this at two levels: empirical evidence of cross-task transfer, and structural generalizability of the formulation.
>
> First, **FlexTravelBench already provides a qualitatively distinct evaluation setting** that tests a fundamentally different planning regime. TravelPlanner is a single-turn itinerary generation task, whereas FlexTravelBench requires **multi-turn constraint adaptation** with progressively revealed constraints. This directly tests the ability to revise prior allocations under evolving requirements — a capability absent in static single-turn planning. HiMAP-Travel still achieves consistent gains of +4.7–5.1 pp over ATLAS across all four scenario types, suggesting the architecture is not overfitted to a single benchmark structure.
>
> Second, the framework’s assumptions are **domain-agnostic**. Section 3 formulates the problem as an HC-POMDP and only requires: (1) **decomposable global constraints**, (2) **decomposable task structure**, and (3) **coupled resource conflicts** among parallel agents. These properties are not specific to travel, so the same Coordinator/Executor/Monitor design can in principle apply to other constrained planning domains. Natural candidates include software project planning (shared developer-hours budget, modular task decomposition), supply-chain optimization (shared inventory, parallel procurement), and scientific experiment design (shared equipment time, independent experimental runs).
>
> We agree that broader cross-domain empirical validation is beyond the scope of the current paper and will clarify this limitation more explicitly.
>
> ---
>
> **W2: Where does the performance gain come from: architectural design or agent reasoning capabilities?**
>
> A: This question is directly addressed by our controlled comparison, and the ablation further clarifies the contribution of individual architectural components.
>
> Our controlled comparison isolates the **architectural contribution** by construction. HiMAP-Travel and the sequential DeepTravel RL baseline use the **same backbone, training algorithm, training data, tool APIs, decoding hyperparameters, and tool-call budget**, so they have the same opportunity to improve reasoning. The resulting gains—**+8.67 pp** for 8B and **+14.88 pp** for 4B, with **93% variance reduction**—therefore come from the architecture rather than stronger reasoning capability alone.
>
> The ablation supports this interpretation. $\Sigma$ does not add reasoning capacity; it only enforces deterministic global invariants such as budget balance and venue uniqueness. Removing it causes a −9.58 pp drop and sharply increases cross-agent conflicts, especially restaurant duplication (7.7% → 34.1%), showing that it prevents coordination failures rather than reasoning for the agent.
>
> More broadly, different ablations produce different failure signatures: removing the **Coordinator** mainly increases budget and cascading failures, while removing the **monitor** mainly increases cross-agent conflicts. This shows the gain comes from architectural mechanisms that structure and safeguard reasoning, not simply from better raw reasoning.
>
> ---
>
> **W3: Re-evaluation of baselines under identical conditions.**
>
> A: We agree that cross-model comparisons should be interpreted cautiously, and we will make this framing even clearer in the revision.
>
> Our **primary architectural claim relies on the controlled comparison, not on cross-model baselines**. Specifically, HiMAP-Travel is compared against a sequential DeepTravel RL baseline under the same backbone, training algorithm, training split, tool APIs, and decoding, so the **+8.67 pp** gain and **93% variance reduction** are the main evidence for the architectural contribution. By contrast, comparisons with *ATLAS* and *MTP* are included as reference points, not controlled evidence of architectural superiority.
>
> Re-evaluating ATLAS or MTP under identical conditions is also not straightforward. *ATLAS* depends on Gemini-2.5-Pro via API, so we cannot reproduce its training pipeline or apply GRPO fine-tuning. *MTP* is built on GPT-4 with manually engineered domain-specific prompts and hints; porting it to Qwen3-8B would require substantial re-engineering and would no longer be a faithful evaluation of the original method.
>
> ---
>
> **Q1: Any other constrained long-horizon agent tasks?**
>
> A: Please see **W1** for the full discussion.

---

> > ### Author Rebuttal · Reviewer_FJUj · 2026-04-02
> >
> > After reading the rebuttal, I decided to maintain the current score.

---

> > > ### Author Response · Authors · 2026-04-04
> > >
> > > Thank you for your acknowledgement. We are pleased that the concerns have been fully addressed and would be happy to answer any further questions. We would also appreciate it if you could kindly reconsider the score in light of this.

---

### Official Review · Reviewer_kcTT · 2026-03-13

**Soundness:** 3
**Presentation:** 3
**Significance:** 2
**Originality:** 3
**Overall Recommendation:** 4
**Confidence:** 2

**Summary:**

HiMAP-Travel addresses constraint drift in sequential LLM planners by decoupling planning into a strategic coordinator and parallel day executors, connected via a transactional global state, a cooperative bargaining protocol, and a single GRPO-trained role-conditioned policy. On TravelPlanner, it achieves 52.6% test FPR with Qwen3-8B, outperforming a controlled sequential baseline by 8.6 pp with 93% variance reduction, and reducing latency in 2.5 times.

**Compliance With Llm Reviewing Policy:**

Affirmed.

**Final Justification:**

The authors presents an architectural solution to constraint drift in sequential LLM planners, and the controlled comparison against DeepTravel isolates the contribution well. The rebuttal addressed my concerns reasonably. The val–test gap explanation via split composition is plausible given stable FPR, the context length analysis is convincing, and I appreciate the honest walk-back on "managerial empathy" to a hypothesis. The budget failure and 8B regression answers were thoughtful. My confidence on this paper is low, so I defer to reviewers with deeper domain expertise, but the work appears solid. I maintain my weak accept.

**Key Questions For Authors:**

1. Are the 31.6% of budget overrun failures shown on table 41 due to poor coordinator allocation or genuinely infeasible queries given the database?
2. Why does the 8B model regress on Room Rule and minimum nights relative to 4B?
3. What prevents extending global state to cover minimum night stay violations, given they represent 13.7% of failures?
Please take a look at the weaknesses section for more questions.

**Limitations:**

Yes.

**Strengths And Weaknesses:**

Strengths
1. Controlled comparison against DeepTravel cleanly isolates the architectural contribution. The 8.6 pp gain and dramatic variance reduction from 7.18% to 0.48% are both impressive.
2. Ablations confirm each component's necessity. Removing the coordinator causes the largest drop (−12.9 pp), followed by global state (−9.5 pp), parallelism (−7.1 pp), and bargaining (−3.8 pp).
3. On appendix F, detailed failure taxonomy with actionable categories is unusually rigorous for an agentic planning paper. That is a great contribution for sure.
4. Honest latency analysis correctly attributes the gap between theoretical 3x and actual 2.6x speedup to fixed overheads.

Weaknesses
1. The Hard Constraint Micro gap of 25 pp between validation and test is large and not fully explained (shown on Table 17). A learning curve over training set size is needed.
2. Table 37 provides only indirect observational evidence. A direct measurement of constraint recall degradation as a function of context length would substantiate the paper's central motivation.
3. The claim that the unified policy benefits from managerial empathy is compelling but unsupported. The ablation only shows the combined effect of separating policies, not the mechanism. I am not convinced about that, but I am open for clarifications.

---

> ### Author Rebuttal · Authors · 2026-03-30
>
> Thank you for the constructive feedback. We respond below to weaknesses (W) and questions (Q).
>
> ---
>
> **W1. The 25 pp val–test gap on Hard Constraint Micro is not well explained; a training-size learning curve is needed.**
>
> A: We believe this gap reflects **distribution shift rather than overfitting**, for three reasons. First, the primary generalization metric is stable: FPR changes by only 0.13 pp from validation (52.78%) to test (52.65%), well within seed variance. Second, Hard Constraint Macro increases on test by +2.72 pp (8B) and +4.28 pp (4B), which is opposite to overfitting. Third, Hard Constraint Micro is highly sensitive to split composition because it aggregates individual hard-constraint checks across queries; if the test set contains more queries activating multiple hard constraints simultaneously, Micro can drop sharply even when per-constraint accuracy is unchanged. Together, the stable FPR and improved Macro suggest this is primarily a split-composition artifact.
>
> ---
>
> **W2. Table 37 gives only indirect evidence; please measure constraint recall vs. context length directly.**
>
> A: In Table 37, we use planning day as a proxy for context length, since in sequential planners, Day *k* conditions on all prior tool outputs and reasoning traces, so context grows monotonically by construction.
>
> Each day adds about 1.5k tokens, giving $\text{context}(k)\approx 800 + k \times 1{,}500.$
> For a 5-day trip, this grows from about **2.3k** tokens (Day 1) to **8.3k** (Day 5), exceeding the **8,192-token** training window and truncating early context, including global constraints. This matches the monotonic degradation in sequential baselines (98% → 42% budget satisfaction), while HiMAP-Travel remains stable because each Executor sees a fresh, fixed-length context.
>
> We will add average accumulated context length alongside constraint satisfaction in the revision.
>
> ---
>
> **W3. The managerial empathy claim is unsupported.**
>
> A: We agree the supported claim is only that a *unified role-conditioned policy helps*; "managerial empathy" was intended as a hypothesis, not a verified mechanism, and we will revise the wording accordingly.
>
> Empirically,  **switching from a unified role-conditioned policy to two independently trained policies reduces FPR by 8.08 pp** (52.78% → 44.70%) under otherwise matched settings, showing a clear benefit of parameter sharing across roles. The ablation does not identify the mechanism: the gain may come from cross-role transfer, shared-reward joint optimization, or regularization. Thus, the effect is supported, but the mechanism remains a hypothesis.
>
> ---
>
> **Q1. Are the 31.6% budget failures due to poor allocation or inherent infeasibility?**
>
> A: Both factors contribute, but the evidence suggests the majority are *inherently infeasible* rather than Coordinator errors. Budget-overrun failures have mean utilization of 107.4% (Table 41), confirming genuine overspending, while the higher bargaining count (2.8 vs. 1.9) shows that many poor initial allocations are corrected during replanning; indeed, 28.6% of failures come from initially infeasible allocations that are recovered through bargaining. By contrast, the 31.6% of budget failures that remain after $K_{max}=3$ iterations are best explained as cases where no valid itinerary satisfies the budget given the database.
>
> ---
>
> **Q2. Why does 8B regress on Room Rule and Minimum Nights versus 4B?**
>
> A: We believe the root cause is **constraint frequency imbalance** in training queries: budget appears in 100% of queries, while room rules (~33%) and others are less frequent. Importantly, **Minimum Nights is not a query-level constraint**—it is implicitly enforced via evaluation, so its signal comes only from the binary FPR reward, without a dedicated component.
>
> Under GRPO, this gives dominant constraints stronger gradients, so 8B model may improve more on frequent constraints while regressing on rarer or implicit ones such as Room Rule and Minimum Nights. The 4B model may generalize slightly better on these niche constraints through more conservative behavior.
>
> Thus, these are local regressions, while 8B still achieves a +2.85 pp gain in overall FPR. We will clarify this limitation in the revision.
>
> ---
>
> **Q3. What prevents extending global state to enforce minimum-night constraints?**
>
> A: Nothing prevents this in principle; we already outline the extension in Appendix *Failure Analysis*. We did not include it because ($\Sigma$) is intentionally restricted to **cross-agent resource conflicts** independent of future trip structure, whereas minimum-night checks depend on the **global city–day assignment**, which may still change during bargaining. Preloading such state before convergence would require reinitialization under `checkpoint/rollback`, increasing complexity and latency. Once the schedule is finalized, the extension is straightforward: propagate it into ($\Sigma$) and enforce the constraint at commit time. We leave this for future work.

---

### Official Review · Reviewer_9aKz · 2026-03-23

**Soundness:** 3
**Presentation:** 2
**Significance:** 2
**Originality:** 2
**Overall Recommendation:** 4
**Confidence:** 4

**Summary:**

The paper addresses the problem of attention drift in long-horizon planning under extended contexts，to this end, it proposes a hierarchical multi-agent planning framework, where a centralized Coordinator decomposes the task and assigns subgoals to multiple sub-agents. These sub-agents plan independently while interacting through a synchronized global state to avoid conflicts. The framework also includes a feedback mechanism to revise infeasible sub-task goals and is further optimized with GRPO.

**Compliance With Llm Reviewing Policy:**

Affirmed.

**Final Justification:**

The detailed response addresses my concerns. I am therefore raising my score to 4.

**Key Questions For Authors:**

1. Regarding the effectiveness of the global constraint mechanism: the paper introduces sub-agent bargaining and a synchronized global state mechanism to prevent conflicts among independently planning agents. In practice, how often do reflection and replanning occur? If this happens too frequently, the framework may suffer from low efficiency.
2. What is the performance of RL-based baselines on FlexTravelBench?
3. In the RL optimization, how is the local reward R_{local} for the lower-level Executors specifically designed and implemented?

**Limitations:**

1. The  effectiveness of the proposed framework.
2. The performance of RL-based baselines on FlexTravelBench.
3. The design of the local reward R_{local} for the lower-level Executors.

**Strengths And Weaknesses:**

# Strength
1. The centralized planner + decentralized actors design is well motivated and can effectively mitigate the issue of context drift in long-horizon planning.
2. The method shows that smaller models can outperform larger ones under this framework, and the experimental results on the benchmarks are generally strong and fairly comprehensive.

# Weakness
1. The method relies heavily on the centralized Coordinator for task decomposition and resource allocation. If the high-level decomposition is inaccurate, errors may propagate down the hierarchy, prematurely restricting the search space of downstream agents and potentially causing repeated replanning or rollback.
2. The reported efficiency gains come with increased compute and token costs. While the paper shows reduced wall-clock latency through parallel execution, its total compute and token usage are in fact higher than those of sequential baselines. In particular, replanning by the high-level planner incurs additional overhead, but the paper does not quantify the frequency of replanning during task execution.
3. In FlexTravelBench, the paper does not compare against RL-based baselines. Such comparisons are important to show that the gains come from the proposed framework itself rather than simply from additional RL fine-tuning.
4. typos:
  - L363 contains garbled text.
  - Figure 2 is missing axis labels in all three subplots.
  - In the appendix (Page 23), one table exceeds the page boundary.

---

> ### Author Rebuttal · Authors · 2026-03-30
>
> We appreciate the reviewer's feedback and provide the following clarifications, using W to denote weaknesses and Q to denote questions.
>
> ---
>
> **W1: Inaccurate high-level planning may propagate down the hierarchy and hurt efficiency.**
>
> A: In HiMAP-Travel, the Coordinator provides soft priors, not hard constraints: per-day budget hints $b_d$ guide planning, while all hard constraints are enforced by the Synchronized Global State $\Sigma$ via atomic commits. Thus, inaccurate high-level planning does not cause failure; Executors can still select any globally valid actions.
>
> When a decomposition is infeasible, the Cooperative Bargaining Protocol triggers: Executors return structured infeasibility signals, the Coordinator revises subgoals, and the system replans from a clean state via `rollback()`. Replanning is capped at $K_{max}=3$, preventing unbounded rollback loops.
>
> Empirically, this mechanism is effective and lightweight: 89.2% of queries succeed in the first iteration, and bargaining adds only 10.8% wall-clock overhead (§5.4). Removing the Coordinator leads to a −12.98 pp drop in FPR, and the low variance (std = 0.48%) indicates stable behavior.
>
> Overall, HiMAP-Travel is robust to imperfect high-level planning, with $\Sigma$ and bargaining serving as explicit safeguards against error propagation.
>
> ---
>
> **W2: Performance gain comes from increased compute and token costs.**
>
> A: We agree that the latency–compute trade-off is important and note that it is already quantified in Table 5. Relative to DeepTravel (8B), HiMAP-Travel (8B) increases compute only modestly (210s vs. 190s, +10.5%) and token usage by about 15% (14.2k vs. 12.3k), while reducing wall-clock latency by 2.63× (72s vs. 190s). This is a favorable trade-off in practice: the cost increase is modest, while the user-facing latency reduction is substantial. Replanning overhead is also small: **89.2%** of queries succeed in the first iteration, replanning is capped at $K_{max}=3$, and the bargaining overhead (**7.8s ± 3.2s**, **10.8%** of wall-clock time for 7-day itineraries) is already included in the **72s** figure.
>
> Overall, HiMAP-Travel’s gains come from a large latency reduction with only modest increases in compute and token usage.
>
> ---
>
> **W3: Did not compare to RL-based baselines on FlexTravelBench.**
>
> A: We address this at two levels.
>
> 1. The **architecture–RL distinction is already controlled on TravelPlanner** (Tables 2 and 4): with identical backbone, RL training, data, tools, and decoding, HiMAP-Travel improves FPR by **+8.67 pp** and reduces variance by **93%**, attributing gains to the framework design.
>
> 2. For **FlexTravelBench**, to our knowledge, no publicly available RL baseline has reported results on this benchmark. We note, however, that FlexTravelBench emphasizes **multi-turn constraint adaptation**, where HiMAP-Travel’s bargaining with `checkpoint/rollback` is naturally suited for structured re-allocation, unlike a sequential RL baseline. The current results already suggest that this architectural advantage persists, e.g., **37.52% vs. 33.60%** in the 3-turn Local→Global setting.
>
> ---
>
> **W4: Typos issues.**
>
> A: We have corrected the typos and improved clarity throughout, and these changes will be included in the revision.
>
> ---
>
> **Q1: How frequently does replanning occur in practice?**
>
> A: Replanning is infrequent in practice. As reported in section 5.4 and Appendix C.1, **89.2% of queries succeed in the first bargaining iteration**, so no replanning is needed, and **97.8% are resolved within 3 iterations**. The associated **overhead is only 7.8s ± 3.2s, only 10.8% of total wall-clock time for 7-day itineraries**. This low rate is also encouraged during training by the Coordinator’s iteration penalty term $-\lambda_{iter} N_{iter}$, which incentivizes feasible sub-goals on the first attempt.
>
> ---
>
> **Q2: Performance of RL-based baselines on FlexTravelBench?**
>
> A: Please see **W3**.
>
> ---
>
> **Q3: How is the $R_{local}$ for the Executors designed and implemented?**
>
> A: We thank the reviewer for this question and will include the full definition of $R_{local}$ in the revision. $R_{local}$ is a **day-level soft-constraint reward** for a single Executor plan $\tau_d$, providing a dense local signal before the global reward:
> $$
> R_{local}(\tau_d)=\sum_{c\in\mathcal{C}_{local}(d)} w_c, s_c(\tau_d)-\eta \max(0,\hat{C}_d-b_d)
> $$
>
> Here, $\mathcal{C}_{local}(d)$ includes day-local constraints (e.g., preferences, room type, commonsense consistency), $s_c(\tau_d)\in[0,1]$ is the satisfaction score, and the penalty term discourages exceeding the Coordinator’s budget hint.
>
> This separates roles: $R_{global}$ evaluates global invariants, while $R_{local}$ provides immediate per-Executor feedback. The training objective is:
> $$
> J_{exec}=\gamma_{global}R_{global}+\gamma_{local}R_{local}+\gamma_{early}\mathbf{1}*{early},
> $$
>
> with $\gamma_{local}=0.3$ and an early-detection bonus $\gamma_{early}=0.15$ to encourage fail-fast behavior.

---

> > ### Author Rebuttal · Reviewer_9aKz · 2026-04-04
> >
> > Thank you for the detailed response which addresses my concerns. I am therefore raising my  score to
> >  4.

---

> > > ### Author Response · Authors · 2026-04-08
> > >
> > > Thank you for your acknowledgement and for raising the score. We appreciate your constructive feedback and would be happy to address any further concerns.

---

### Decision · Program_Chairs · 2026-04-30

**Decision:**

Accept (regular)

**Comment:**

The paper studies the problem of long-horizon (multiple-day) travel planning under hard (e.g. budget) and soft global constraints. The proposed approach employs hierarchical planning: a centralized coordinator decomposes the task into subgoals (‘days’) and soft constraints, while parallel executers independently try to achieve their assigned sub-goals. To ensure feasibility, executers may reject infeasible goals via a cooperative bargaining protocol. The executers interact through a synchronized global state that enforces constraint satisfaction (such as budget constraints and no double bookings). The method is evaluated on the TravelPlanner and FlexTravelBench benchmarks.

The reviewers generally agreed that the paper identifies a genuine issue with sequential planners – propagation and enhancement of early errors - and that the proposed solution is well motivated. They also found the experimental evaluation thorough and the performance gains substantial. In particular, they appreciated the ablation study and the controlled comparison against the DeepTravel baseline.

On the other hand, the reviewers also raised several concerns. First, some of the comparisons were done versus implementations based on older backbone models, which may make the comparison less fair.  Additional concerns included the relatively small dataset size and the restriction of the experiments to the travel-planning domain, (as well as several other issues that were largely addressed in the authors’ response).

Overall, I believe that most of these concerns were adequately addressed. Given the paper’s algorithmic and empirical contributions, I recommend acceptance.